# Bioinspired artificial antioxidases for efficient redox homeostasis and maxillofacial bone regeneration

Ting Wang[1,5], Mingru Bai[2,3,5], Wei Geng [1], Mohsen Adeli [4], Ling Ye [2,3] & Chong Cheng [1,2] ✉

Reconstructing large, inflammatory maxillofacial defects using stem cell-based therapy faces challenges from adverse microenvironments, including high levels of reactive oxygen species (ROS), inadequate oxygen, and intensive inflammation. Here, inspired by the reaction mechanisms of intracellular antioxidant defense systems, we propose the de novo design of an artificial antioxidase using Ru-doped layered double hydroxide (Ru-hydroxide) for efficient redox homeostasis and maxillofacial bone regeneration. Our studies demonstrate that Ru-hydroxide consists hydroxyls-synergistic monoatomic Ru centers, which efficiently react with oxygen species and collaborate with hydroxyls for rapid proton and electron transfer, thus exhibiting efficient, broad-spectrum, and robust ROS scavenging performance. Moreover, Ru-hydroxide can effectively sustain stem cell viability and osteogenic differ-entiation in elevated ROS environments, modulating the inflammatory microenvironment during bone tissue regeneration in male mice. We believe this Ru-hydroxide development offers a promising avenue for designing antioxidase-like materials to treat various inflammation-associated disorders, including arthritis, diabetic wounds, enteritis, and bone fractures.

Maxillofacial bone defects that include the maxilla and mandible arising from degenerative diseases, trauma, congenital anomalies, or tumors can lead to significant functional impairments and significantly diminish patients' life quality[1,2], which affects about 2 million patients all over the world. Because of their high inflammation and large defect volumes, maxillofacial bone defects are difficult to self-repair[3–5], which requires extra clinical management, such as the transplantation of autologous/allogeneic or artificial bone grafts[6,7]. However, this approach has been confronted with several drawbacks, including infections, immune rejection, donor scarcity, and suboptimal osseointegration, often necessitating secondary surgical procedures due to issues like heterogeneous scar formation, delayed healing, or nonunion[8,9]. Current approaches employing biomineralized hydrogels also encounter some critical challenges in matching the required regeneration speed, mechanical strength, redox homeostasis, and anti-inflammatory capability, thereby significantly limiting their applicability in the regeneration of inflammatory bone defects[10–12]. The above challenges of current strategies for repairing large and inflam-matory maxillofacial defects have led to increasing interest in the development of stem cell-based therapy[13–16].

The tissue regeneration in maxillofacial bone defects primarily occurs through intramembranous osteogenesis, where endogenous stem cells differentiate into osteoblasts and subsequently form intri-cate bone structures[17,18]. Establishing a conducive microenvironment

[1]College of Polymer Science and Engineering, State Key Laboratory of Polymer Materials Engineering, Sichuan University, Chengdu, China. [2]State Key Laboratory of Oral Diseases, National Center for Stomatology, National Clinical Research Center for Oral Diseases, West China Hospital of Stomatology, Sichuan University, Chengdu, China. [3]Department of Cariology and Endodontics, West China Hospital of Stomatology, Sichuan University, Chengdu, China. [4]Institute of Chemistry and Biochemistry, Free University of Berlin, Berlin, Germany. [5]These authors contributed equally: Ting Wang, Mingru Bai. ✉e-mail: chong.cheng@scu.edu.cn

for stem cell infiltration and differentiation is crucial in addressing inflammatory maxillofacial defects[19,20]. However, the therapeutic effectiveness of endogenous stem cell infiltration is often hindered by the presence of accumulated reactive oxygen species (ROS) and inadequate oxygen levels[21,22]. Particularly in inflammatory bone defects, elevated ROS levels can exacerbate local tissue damage and trigger chronic inflammation[23–27]. Hence, the removal of excess ROS to create a favorable microenvironment for endogenous stem cells has emerged as a promising strategy to enhance bone tissue regeneration in the management of inflammatory bone defects[28–31]. Currently, chemists and material scientists are deeply involved in the research and development of biocatalytic materials to serve as artificial antioxidases, including metal oxides, nanocarbons, and metallic nanoparticles, aimed at enhancing the ROS elimination activity to safeguard redox homeostasis and support the viability of stem cells[32–36]. Nevertheless, conventional artificial antioxidases exhibit sluggish enzymatic reaction rates due to the deficiency of proton/electron transfer within the active centers[37,38]. Moreover, the susceptibility of biocatalytic centers to ROS-induced poisoning hampers the long-term ROS elimination activity due to challenges in creating high electron density and achieving fast electron recovery[39–41]. Therefore, it is a significant

challenge to construct efficient and durable artificial antioxidases via advanced strategies for ROS clearance to protect endogenous stem cells and promote bone regeneration in inflammatory maxillofacial defects.

In bioorganisms, natural enzymes usually do not operate independently, but rather, a series of enzymes participate in the same biological process[42–44]. For instance, the intracellular antioxidant defense systems (IADS) require the collaborative efforts of glutathione peroxidase (GPx), superoxide dismutase (SOD), catalase (CAT), and others to uphold the cellular redox balance[45,46]. The effective ROS elimination by the IADS involves rapid proton and electron transfer processes and also hydrogen contained substrates or coenzymes (Fig. 1a)[47,48]. For example, the biocatalytic removal of $H_2O_2$ by GPx depends on the rapid deprotonation of the hydrogen-contained glutathione (GSH) substrate, followed by the nicotinamide adenine dinucleotide (NADH)-assisted reprotonation process[49–52]. Similarly, the biocatalytic elimination of superoxide anions ($\bullet O_2^-$) via SOD also requires to undergo electron transfer and protonation processes[53–56]. Consequently, the development of artificial antioxidases with superior catalytic reaction kinetics necessitates a meticulous emulation of the swift proton and electron transfer mechanisms observed in natural

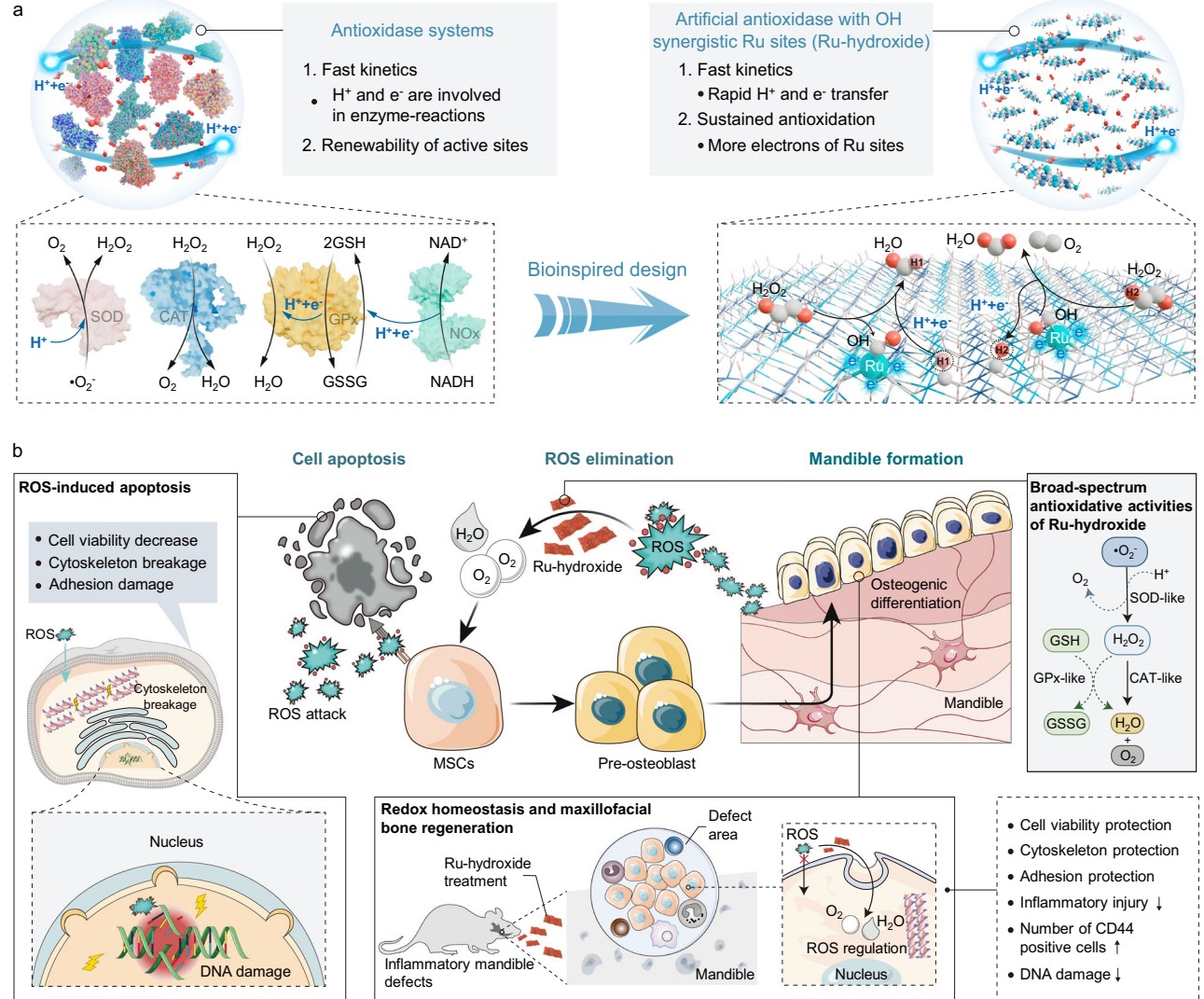

**Fig. 1 | Bioinspired design of artificial antioxidases and therapeutic effects in treating inflammatory maxillofacial bone defects. a** Illustration of the bioinspired design of Ru-hydroxide with rapid and sustained protons/electrons transfer functions, and (**b**) it serves as the antioxidase-like biocatalytic materials for protecting endogenous stem cells and promoting inflammatory mandible regeneration.

IADS, which encompasses not only the deprotonation phase but also the reprotonation process to ensure rapid and sustained biocatalytic redox reaction. Evolution has led to the development of enzymes that typically incorporate the transition metal species, such as iron (Fe), copper, and manganese[57,58]. Ruthenium (Ru), as a member of iron group elements, presents advantageous biocompatibility and unique metallic properties[59,60]. In comparison to Fe species, the Ru features more *d*-electrons, sufficient unoccupied orbitals, and superior redox stability, which benefit bond formation, reactive intermediate coordination, and adsorption-dissociation dynamics during catalysis[61,62]. Therefore, constructing Ru-based artificial antioxidases with rapid proton and electron transfer mechanisms presents a promising avenue for enhancing the catalytic ROS-elimination activities and supporting redox homeostasis.

Here, to mimic this natural IADS mechanism for efficient redox reaction, we propose the de novo bioinspired design of an efficient artificial antioxidase via using Ru-doped layered double hydroxide (named Ru-hydroxide) for superior redox homeostasis and maxillofacial bone regeneration. The main motivation of our research originates from three aspects (Fig. 1a, b): (1) the innovative hydroxyls-synergistic monoatomic Ru centers in Ru-hydroxide enable rapid and sustained proton/electron transfer for efficient ROS biocatalysis; (2) the high electron density of the Ru centers optimizes the binding strength of oxygen species and enhances the resistance to $H_2O_2$ toxicity for Ru-hydroxide; (3) Ru-hydroxide can maintain superior redox balance for efficiently protecting endogenous stem cells, inhibiting inflammatory responses, and regenerating mandibular bone. Moreover, our studies validate that the Ru catalytic sites can react with oxygen species efficiently and cooperate with hydroxyls to complete rapid proton and electron transfer, thus significantly reducing the activation energy of antioxidative reactions. Accordingly, the Ru-hydroxide exhibits efficient and simultaneous SOD-, CAT-, and GPx-mimetic activities for ROS elimination, as well as favorable cycling stability for long-term usage. The resulting Ru-hydroxide demonstrates the antioxidase-like ability to sustain stem cell viability and promote osteoblastic differentiation under conditions of elevated ROS. These biomimetic artificial antioxidases effectively mitigate oxidative stress-mediated DNA damage and cellular apoptosis, thus supporting critical metabolic pathways, developmental cascades, and osteogenic functionality while attenuating pro-inflammatory signaling mechanisms. Strikingly, the synthesized Ru-hydroxide exhibits efficient in vivo biocatalytic microenvironment modulation during inflammatory bone tissue regeneration, providing a hopeful avenue for developing antioxidase-like biomaterials to treat a broad range of inflammation-associated diseases, such as arthritis, diabetic wounds, enteritis, and bone fractures.

## Results

### Synthesis and characterization of artificial antioxidases

The bioinspired Ru-hydroxide with monoatomic Ru and hydroxyl co-catalytic sites was synthesized through a facile two-step hydrothermal method (Fig. 2a and Supplementary Figs. 1, 2, See Methods for details). In brief, the CoNi layered double hydroxide (LDH) precursor (named hydroxide) was prepared first, and then $RuCl_3$ was added to react with the hydroxide. And the collected sample was named Ru-hydroxide. Here, the Ru atoms can be anchored by the plentiful -OH end groups on the hydroxide. Samples with different Ru contents anchored on hydroxide were also prepared. Given that Ru-hydroxide with a ratio of 1:5 exhibits optimal catalytic dynamics in ROS scavenging, thus it should be noted that all references to Ru-hydroxide samples refer to Ru-hydroxide (1:5) unless otherwise specified. To disclose the effects of OH groups on ROS catalysis, the hydroxide was subjected to dehydrogenation in an oxygen environment at a temperature of 400 °C to produce the CoNi oxide ($CoNiO_x$) without surface H atoms (named oxide). Thereafter, the control sample, monoatomic Ru doped

oxide (named Ru-oxide) with similar Ru contents, was also prepared as described in the Methods section (Fig. 2b).

The crystalline phases are identified by X-ray diffraction, confirming that Ru-hydroxide has a hydrotalcite-like structure and Ru-oxide has a spinel-like structure (Supplementary Fig. 3). The scanning electron microscopy (SEM) images reveal a submicron sheet morphology for Ru-hydroxide and a cracked sheet morphology for Ru-oxide (Supplementary Figs. 1, 4). Transmission electron microscopy (TEM) images validate these morphologies (Supplementary Fig. 5), and the energy dispersive X-ray spectroscopy (EDS) images indicate a uniform distribution of Ru, Co, Ni, and O in both Ru-hydroxide and Ru-oxide, while aberration-corrected high-angle annular dark-field scanning TEM (AC-HAADF-STEM) images prove that the distribution of Ru can be found on the atomic scale (Fig. 2c,d, and Supplementary Figs. 6, 7).

The surface electronic structures and valence states are analyzed by X-ray photoelectron spectroscopy (XPS; Supplementary Figs. 8, 9 and Supplementary Table 1). The peaks at 462.59 eV and 484.78 eV, attributed to Ru $3p_{3/2}$ and Ru $3p_{1/2}$ spin-orbit splitting, suggest the $Ru^{n+}$ oxidation state in Ru-hydroxide and no Ru metallic state (Fig. 2e). Similarly, Ru atoms in Ru-oxide maintain a positive valence state as well. Especially, Ru $3p$ peaks of Ru-hydroxide shift downward by 0.79 eV compared to Ru-oxide, while the Co $2p$ peaks and Ni $2p$ peaks of Ru-hydroxide shift to a higher field compared to hydroxide (Supplementary Fig. 10). These results imply a significant transfer of electrons from the hydroxide to the Ru atoms, leading to Ru sites with a high electron density for efficient ROS biocatalysis. Besides, the O $1s$ binding energy of Ru-hydroxide shows a higher hydroxyl peak than Ru-oxide (Fig. 2f). The hydroxyl groups of Ru-hydroxide have also been confirmed by Raman spectra with the existence of unique M-O peaks belonging to metal hydroxides (Fig. 2g).

Subsequently, to elucidate the accurate coordination structures of Ru sites in Ru-hydroxide, we employ X-ray absorption near-edge structure (XANES) and extended X-ray absorption fine structure (EXAFS) spectroscopy. The XANES spectra of the Ru $K$-edge prove that the Ru-hydroxide exhibits a lower absorption edge position compared to the pristine $RuCl_3$ salts, possessing an average valence state of +1.97 (Fig. 2h, i). This observation indicates a transfer of electrons from the hydroxide to the Ru species, corroborating our XPS results. The Fourier transform of the $k^2$-weighted EXAFS spectra reveals a distinct peak at ~1.5 Å for Ru-hydroxide, which corresponds to the Ru-O bond (Fig. 2j). Additional support for the presence of the Ru-O bond in Ru-hydroxide, with an average coordination number of 3.15, is provided by wavelet-transform (WT) images and EXAFS fitting results (Fig. 2k, Supplementary Fig. 11, and Supplementary Table 2). These findings further validate the absence of metallic Ru species in the Ru-hydroxide biocatalytic materials.

### Antioxidase-like performance evaluation and theoretical analysis

Having effectively elucidated the structure and coordination environments of Ru-hydroxide, particularly focusing on its monomeric Ru and hydroxyl co-catalytic centers, we proceeded to evaluate and contrast its antioxidase-like performance in terms of ROS elimination. Primarily, the biocatalytic conversion of $\bullet O_2^-$ (SOD-mimics) into $H_2O_2$ and $O_2$ by Ru-hydroxide was studied, which demonstrates its superior efficacy in $\bullet O_2^-$ elimination compared to Ru-oxide, hydroxide, and oxide (Fig. 3a). In-depth studies have been performed to assess the long-term activity and stability of Ru-hydroxide as a SOD-like antioxidase; the data indicates that Ru-hydroxide can sustain its high activity with no apparent reduction in performance over the course of six cycles (Supplementary Fig. 12). Moreover, Ru-hydroxide demonstrates an efficient ability to reduce the $H_2O_2$ concentration by ~86% within 12 min, significantly surpassing the performance of Ru-oxide, hydroxide, and oxide (Fig. 3b, c, and Supplementary Fig. 13).

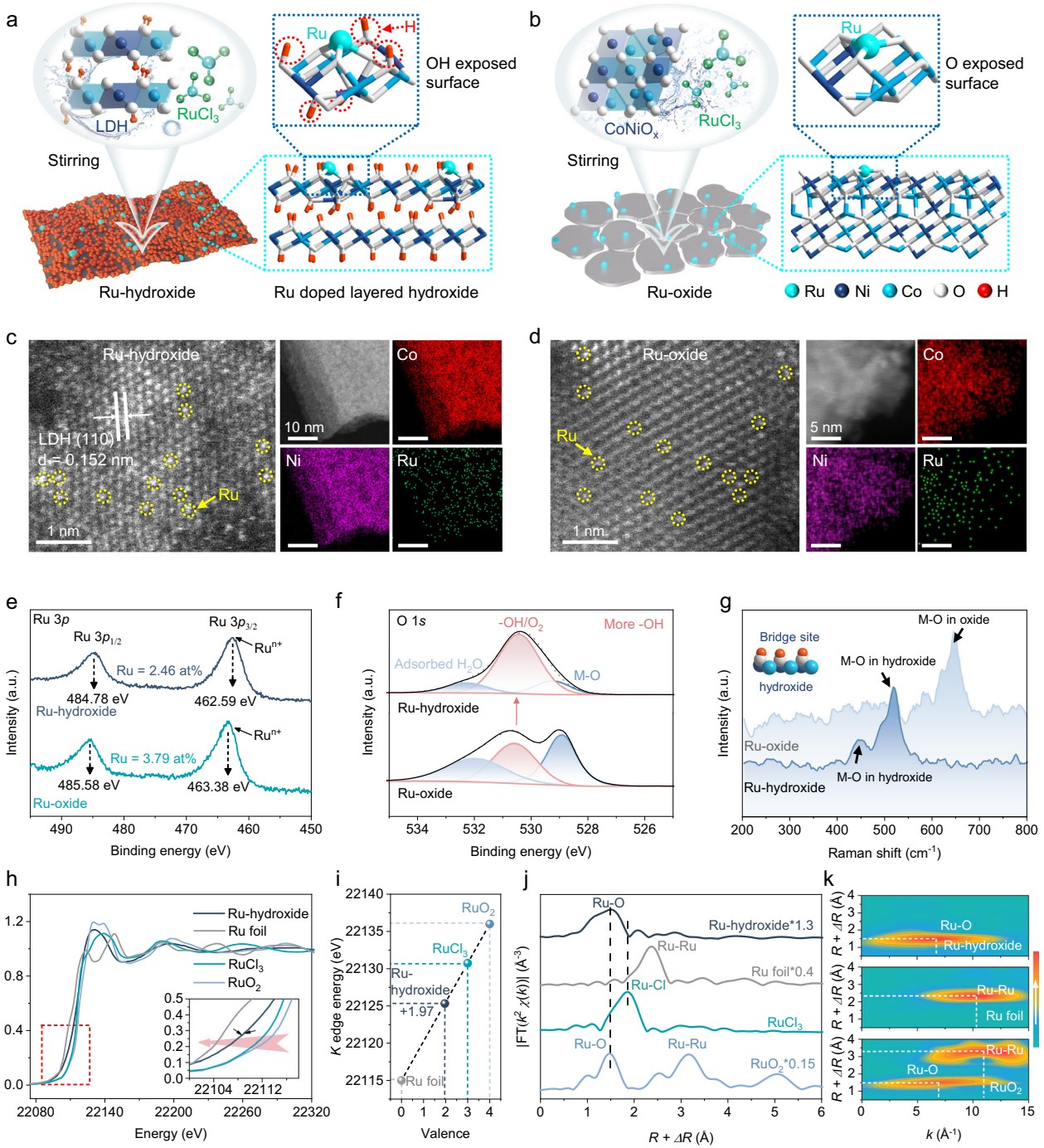

**Fig. 2 | Preparation and structure characterization of antioxidase-like biocatalytic materials.** The synthesis processes and crystal structures of (**a**) Ru-hydroxide and (**b**) Ru-oxide. Atom colors: cyan, Ru; white, O; pink, H; wathet blue, Co; and navy blue, Ni. AC-HAADF-STEM images (Ru atoms are marked by yellow circles), HAADF-STEM images, and the corresponding elemental mapping images of (**c**) Ru-hydroxide and (**d**) Ru-oxide. High-resolution XPS of (**e**) Ru $3p$ and (**f**) O $1s$ of Ru-hydroxide and Ru-oxide, M-O indicates the metal-O bond. **g** Raman spectra of

Ru-hydroxide and Ru-oxide. Atom colors: white, O; pink, H; wathet blue, Co; and navy blue, Ni. **h** Ru $K$-edge XANES spectra, (**i**) Ru oxidation state, and (**j**) Fourier-transformed $k^2$-weighted EXAFS spectra of Ru-hydroxide and standard samples (Ru foil, RuCl₃, and RuO₂). **k** WT analysis at the Ru $K$-edge of different samples. Experiments were repeated independently (**c**–**g**) three times with similar results. In (**e**–**h**), a.u. indicates the arbitrary units. Source data are provided as a Source Data file.

Furthermore, the production of $O_2$ in the CAT-like reaction displays analogous trends, with Ru-hydroxide demonstrating a superior rate of $O_2$ generation, significantly outpacing Ru-oxide and other references (Fig. 3c).

Additionally, the CAT-like reaction kinetics for $O_2$ production have also been investigated by studying the correlation between substrate concentrations and reaction rates. The Ru-hydroxide

demonstrates a notably higher $K_m$ compared to Ru-oxide, suggesting superior desorption capabilities on intermediates, which aligns with the results of our density functional theory (DFT) calculations (Fig. 3d, Supplementary Fig. 14, and Supplementary Table 3). The maximal reaction velocity ($V_{max}$), a key parameter in steady-state kinetics determined through Michaelis-Menten fitting (Supplementary Figs. 15, 16), reveals that Ru-hydroxide exhibits a significantly higher $V_{max}$ value

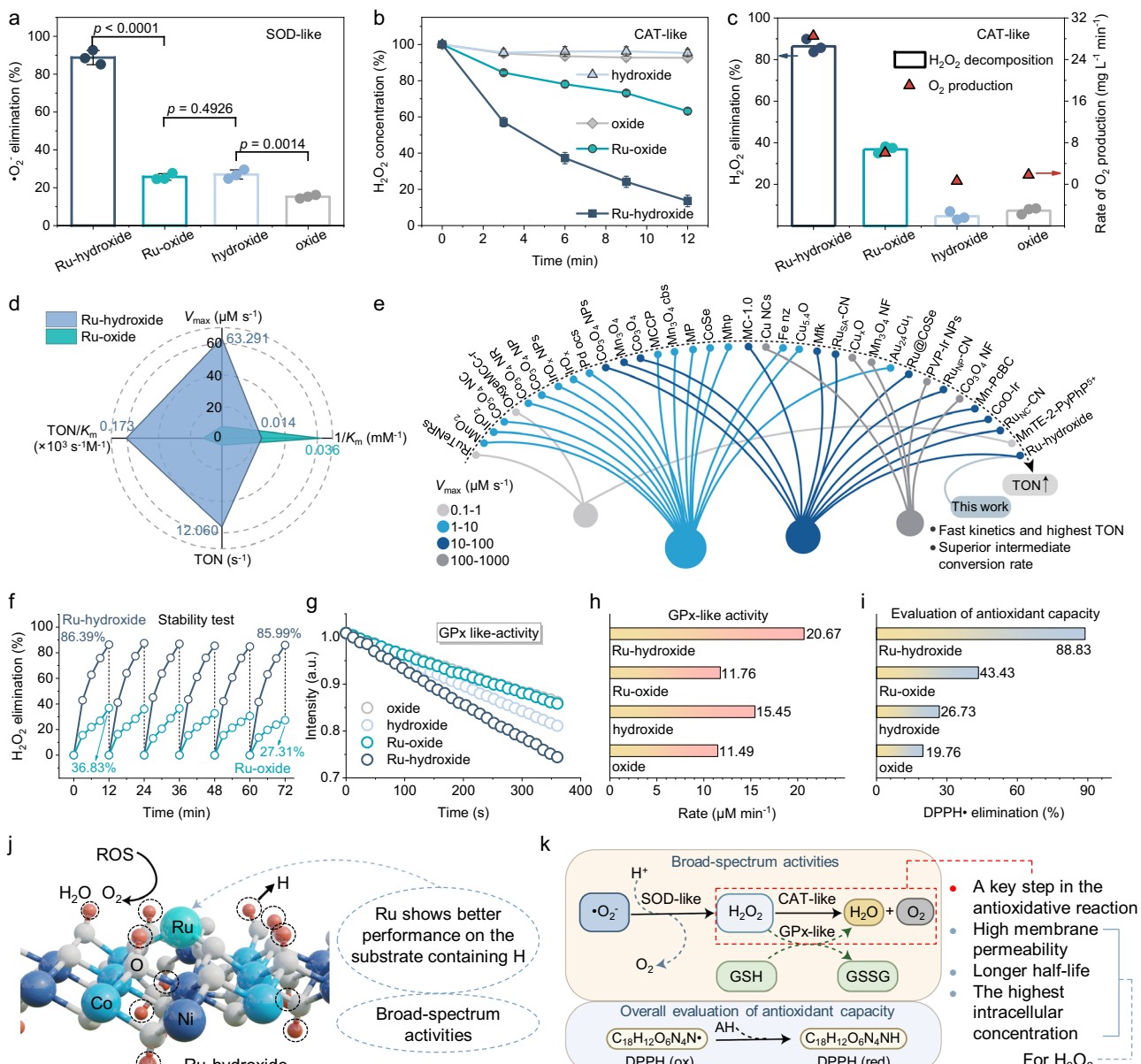

**Fig. 3 | Evaluation of antioxidase-like activities. a** SOD-like activity of Ru-hydroxide, Ru-oxide, hydroxide, and oxide ($n = 3$ independent experiments, data are presented as mean values ± SD). **b** Time-dependent CAT-like activity via TiSO$_4$-based UV-vis spectra in the presence of biocatalytic materials and H$_2$O$_2$ ($n = 3$ independent experiments, data are presented as mean values ± SD). **c** H$_2$O$_2$ elimination ratio of different catalysts ($n = 3$ independent experiments, data are presented as mean values ± SD). **d** $V_{max}$, $1/K_m$, TON, and TON/$K_m$ values of Ru-hydroxide and Ru-oxide. **e** Comparative analysis of the $V_{max}$ (color dots) and TON (arrows) with previously reported biocatalytic materials. **f** Stability test of Ru-hydroxide and Ru-oxide to eliminate H$_2$O$_2$. **g** Time-dependent GPx-like activity, (**h**)

GPx-like reaction rate, and (**i**) DPPH• elimination ratio of Ru-hydroxide, Ru-oxide, hydroxide, and oxide. **j** Ru-hydroxide with surface H from OH group shows better ROS elimination activities than Ru-oxide without surface H. Atom colors: cyan, Ru; white, O; pink, H; wathet blue, Co; and navy blue, Ni. **k** Schematic diagram of ROS elimination through broad-spectrum antioxidative activities of Ru-hydroxide, $V_{max}$ is the maximal reaction velocity, $K_m$ is the Michaelis constant, and TON is the turnover number. In (**g**), a.u. indicates the arbitrary units. In (**a**), statistical significance was calculated using two-tailed Student's $t$-test, all tests were two-sided. Source data are provided as a Source Data file.

of 63.29 μM s$^{-1}$, surpassing Ru-oxide by 8.48 times. When normalizing for the number of Ru sites, the turnover number (TON) of Ru-hydroxide is calculated to be 12.06 s$^{-1}$, exceeding that of Ru-oxide by 12.70 times. Notably, among the current CAT mimics, Ru-hydroxide displays the highest TON values as an antioxidase-mimetic biocatalytic material (Fig. 3e and Supplementary Table 4). The long-term activity and stability of Ru-hydroxide to serve as a CAT-like antioxidase have also been investigated in detail; the data shows that Ru-hydroxide can maintain high activity with no obvious performance decline after 6 cycles (Fig. 3f).

Besides the SOD-like and CAT-like antioxidase performance, we have conducted an investigation into the GPx-like activity. Our findings reveal that Ru-hydroxide exhibits the fastest GPx-like reaction rate among various biocatalysts (Fig. 3g, h). Moreover, a comprehensive assessment of the antioxidant activities of the materials was carried out through the 1,1-diphenyl-2-picrylhydrazyl radical (DPPH•) test, confirming that Ru-hydroxide possesses rapid scavenging activity against various free radicals (Fig. 3i). The above results suggest that the H-rich surface of Ru-hydroxide with monoatomic Ru and hydroxyl sites shows superior, broad-spectrum, enduring, and cascade

antioxidase-like capabilities compared to Ru-oxide, indicating that H atoms (or OH groups) in Ru-hydroxide have a significantly positive influence on the catalytic performance of Ru sites (Fig. 3j, k).

After confirming the superior antioxidase-like activities of Ru-hydroxide, we then carry out the theoretical analysis to disclose the mechanisms of their biocatalytic activities and pathways. $H_2O_2$ holds significant relevance among biologically pertinent ROS due to its elevated intracellular concentration, longer half-life compared to other ROS, and ability to penetrate cell membranes[39,63]. Moreover, the decomposition of $H_2O_2$ is a critical step in cascade antioxidase-like reactions, and natural enzymes, like CAT and GPx, are involved in this process. Therefore, we primarily focus on understanding the mechanisms of catalysts in the decomposition of $H_2O_2$. Primarily, the reactivity of various metal sites in Ru-hydroxide was assessed through Hirshfeld charge analysis. The results indicate that Ru centers exhibit higher electrophilicity, with an electron density of −0.542 |e|, in comparison to Co sites (−0.198 |e|) and Ni sites (−0.180 |e|), suggesting that Ru centers are more inclined to interact with oxygen species during catalysis (Supplementary Fig. 17). Furthermore, the Hirshfeld charges of Ru sites for the Ru-hydroxide and Ru-oxide are studied, revealing that the Ru sites in Ru-hydroxide lose fewer electrons (−0.542 |e|) compared to Ru-oxide, indicating a significantly higher electron density in Ru-hydroxide (Fig. 4a, b).

Subsequently, computational analyses utilizing DFT were conducted to elucidate the fundamental relationship between the hydroxyl-synergistic Ru centers and their catalytic efficacy in $H_2O_2$ decomposition (Supplementary Data 1–3). The pathways governing the $H_2O_2$ decomposition reaction are comprehensively illustrated in Fig. 4c and Supplementary Figs. 18–21. Our findings suggest a hydroxyl-assisted deprotonation and reprotonation pathway, wherein the rapid transfer of H and electrons between Ru-hydroxide occurs, leading to a low energy barrier for the rate-determining steps (RDS) of about 0.14 eV for Ru-hydroxide (Fig. 4d). Furthermore, the Hirshfeld charge analysis illustrated in Supplementary Fig. 22 indicates that, aided by the presence of lattice hydrogen, the Ru center in Ru-hydroxide exhibits significant electron transfer and rapid valence transitions throughout the catalytic process. It is noteworthy that the conversion of the *OOH intermediate to generate *OO and *H remains a high-energy barrier in Ru-oxide, whereas it proceeds spontaneously in Ru-hydroxide due to the reprotonation process of the hydroxide substrate (Fig. 4d). In this process, the substantial attraction of the H vacancy on Ru-hydroxide facilitates the return of H from the *OOH intermediate to Ru-hydroxide, resulting in the production of *OO and ultimately leading to the release of $O_2$ (Supplementary Fig. 20). Consequently, we then take *OOH as a pivotal and representative model to thoroughly explore the fundamental connection between hydroxyls-synergistic Ru centers and biocatalytic characteristics.

The partial density of states (PDOS) peaks observed before and after the adsorption of *OOH indicate that the orbitals of Ru in Ru-hydroxide undergo alterations, suggesting a critical interaction between Ru and *OOH (Supplementary Fig. 23). Moreover, as presented in Fig. 4e, f, the analysis of Bader charge and PDOS reveal that the Ru center in Ru-hydroxide, with a higher charge density, has a weaker interaction with the *OOH intermediate. Additionally, the electrostatic potential (ESP) in Fig. 4g indicates that the H atom on *OOH engages in electrostatic interactions with the H vacancy on Ru-hydroxide. This interaction is conducive to the reprotonation process of H vacancy and significantly reduces the energy barriers for *OOH decomposition and desorption. In contrast, due to the vertical distribution of the *OOH intermediate at the Ru site on Ru-oxide, the H atom on *OOH is unfavorable for the interaction with the Ru site, resulting in unfavorable decomposition of *OOH on Ru-oxide (Fig. 4g).

The hydroxyl-assisted deprotonation and reprotonation pathways were further verified by experiments. As shown in electrochemical cyclic voltammetry (CV) curves, the Ru-hydroxide exhibits obvious peaks of underpotential deposition hydrogen, thus indicating that the H* adsorption and H* desorption amounts on Ru-hydroxide are much larger than Ru-oxide (Fig. 4h). In-situ Fourier Transform infrared (FTIR) spectra also demonstrate the production of OOH (1226 cm⁻¹) and OH (1319 cm⁻¹) species during the decomposition of $H_2O_2$, mirroring the DFT-calculated H-transfer processes (Fig. 4i, j)[64–66]. These results indicate that Ru-hydroxide exhibits significantly lower energy barriers for $H_2O_2$ decomposition reaction due to its rapid H and electrons transfer when compared to the Ru-oxide, which allows the Ru-hydroxide to achieve efficient and stable antioxidase-like activities.

## In vitro ROS damage defense of stem cells via Ru-hydroxide

After confirmation of the effective and broad-spectrum antioxidase-like activities of Ru-hydroxide, we subsequently undertake a comprehensive evaluation of its protective capacity against ROS-induced damage, employing human mesenchymal stem cells (hMSCs) as a model system. Intracellular ROS levels are explored using the fluorescence probe 2,7-dichlorodihydrofluorescein diacetate (DCFH-DA), as depicted in Fig. 5a–c and Supplementary Fig. 24. Upon exposure of hMSCs to $H_2O_2$ (100 μM), a significant increase in fluorescence intensity (green) indicative of high intracellular ROS levels is observed. Treatment with Ru-hydroxide leads to a marked attenuation of the green fluorescence signals, whereas the control sample (Ru-oxide) exhibits only a slight decrease in the signal. And there is no significant reduction in ROS levels for the bare oxide and hydroxide groups. Notably, because the cell metabolism also produces $H_2O_2$, the MSCs +$H_2O_2$+Ru-hydroxide can maintain a relatively stable concentration of ~0.38 μM $H_2O_2$, which is below the 1 μM threshold and comparable to the level in the bare MSCs group without any treatment. This suggests that the Ru-hydroxide treatment may not adversely affect the $H_2O_2$-related self-renewal capability of the stem cells (Supplementary Fig. 25)[67]. In addition to stem cells, osteoblasts and dental pulp cells are also present in the tissue surrounding the damaged tooth root, and the retention of these cells benefits tissue repair. As illustrated in Supplementary Figs. 26, 27, Ru-hydroxide also shows efficient defense capability on ROS damage in MC3T3 (a pre-osteoblast cell line) and human dental pulp cells. Moreover, the live/dead cell is calculated by incubating hMSCs Ru-hydroxide for 24 h, which reveals no major difference between Ru-hydroxide and the Ctrl group, indicating high biocompatibility of Ru-hydroxide (Supplementary Fig. 28). Notably, Fig. 5d demonstrates that Ru-hydroxide can efficiently support stem cell survival, as evidenced by a decrease in the proportion of dead cells.

Furthermore, transcriptomic analysis of hMSCs cultured under normal conditions or $H_2O_2$ exposure is performed to investigate the impact of elevated ROS levels on cellular behaviors. Kyoto Encyclopedia of Genes and Genomes (KEGG) classification of cellular processes between the control (Ctrl) and $H_2O_2$-treated groups reveals alterations in focal adhesion and cytoskeletal architecture in high ROS microenvironments (Fig. 5e). Thereafter, stem cell spreading and adhesion are further evaluated under elevated ROS conditions to elucidate the protective effects of Ru-hydroxide. As shown in Fig. 5f, the stem cells exhibit diminished spreading capacity, cellular retraction, and disorganized cytoskeletal architecture in the high ROS condition. In contrast, Ru-hydroxide-treated hMSCs subjected to $H_2O_2$ maintain their characteristic polygonal morphology, exhibiting pronounced elongation and well-defined F-actin filament organization (gray) circumscribing the nuclear region (blue). The focal adhesion protein paxillin, crucial for adhesion complex assembly and signal transduction, displays immature and diminished focal contact formation under high ROS conditions ($H_2O_2$ group), as shown in Fig. 5f. Notably, Ru-hydroxide addition maintains paxillin expression patterns, characterized by mature focal adhesions and diffuse distribution along cellular peripheries. Quantitative analysis of cellular surface area (Fig. 5g) and paxillin plaque area (Fig. 5h) further substantiates the superior cytoprotective efficacy of Ru-hydroxide compared to bare

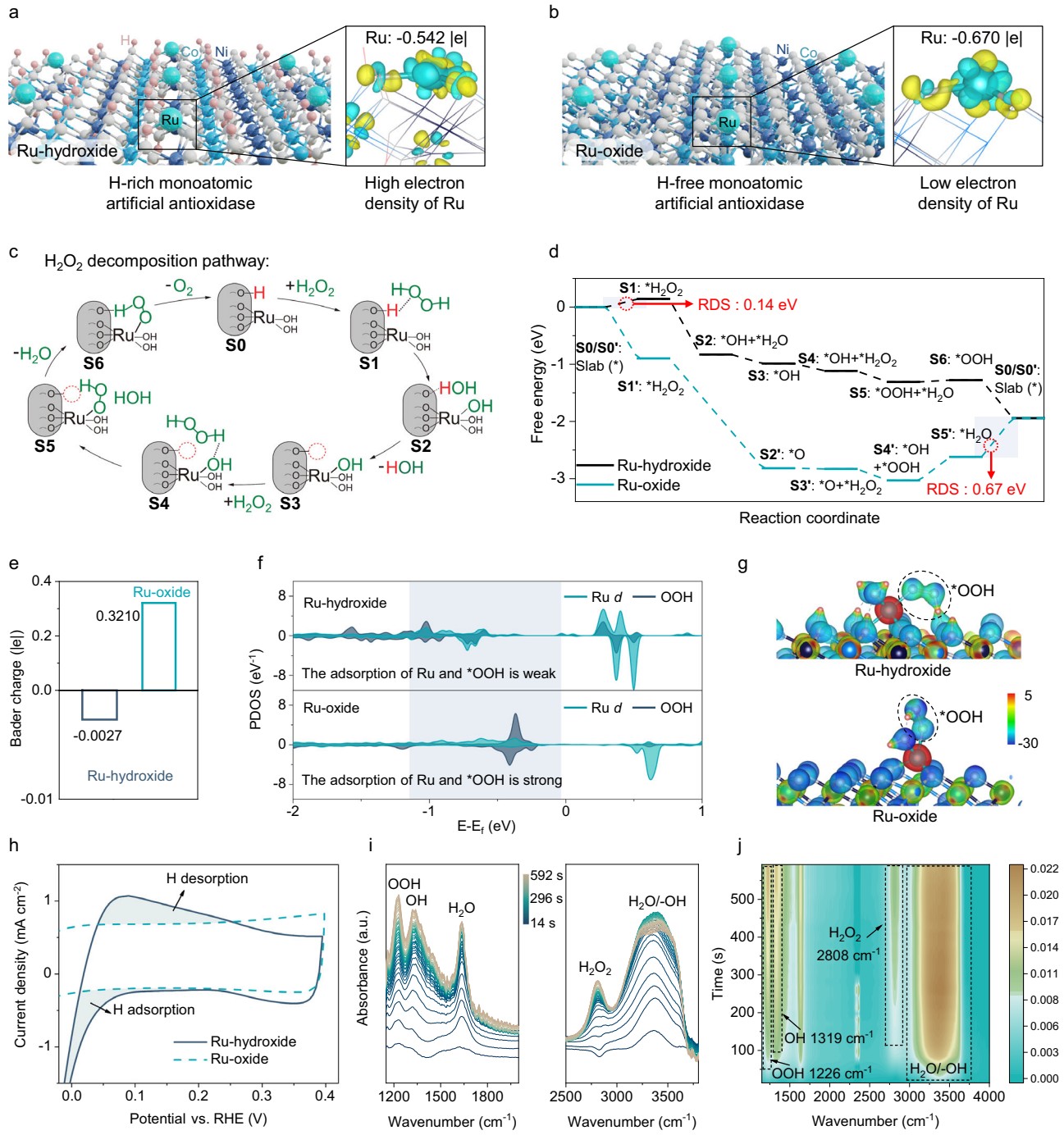

**Fig. 4 | Theoretical analysis to disclose the mechanisms for superior antioxidase-like activities.** Differential charge density analysis and Hirshfeld charge of Ru centers in (**a**) Ru-hydroxide and (**b**) Ru-oxide (yellow and cyan represent charge accumulation and depletion, respectively, with a cutoff value of 0.006 e·Bohr⁻³ for the density-difference isosurface). Atom colors: cyan, Ru; white, O; pink, H; wathet blue, Co; and navy blue, Ni. **c** Proposed hydroxyl-assisted H transfer mechanism of $H_2O_2$ decomposition pathway on Ru-hydroxide. **d** Gibbs free-energy diagram of $H_2O_2$ decomposition on Ru-hydroxide and Ru-oxide. **e** Calculated Bader charge of Ru-hydroxide and Ru-oxide with the desorption of a

*OOH intermediate. **f** PDOS (where Ru corresponds to the *d* orbital, and OOH is the superposition of *s* and *p* orbitals for O and H) and (**g**) electrostatic potential (ESP) analysis of Ru-hydroxide and Ru-oxide with the adsorption of an *OOH intermediate, ESP-mapped surface charge density with the isosurface of 0.2 e·Bohr⁻³. The color scale bar is shown at the right, while the corresponding ESP values (units of eV) from −30 to −5. **h** The CV curves of Ru-hydroxide and Ru-oxide performed between 0 V and 0.4 V in the Ar-saturated 0.1 M PBS. **i** In-situ FTIR spectrum and (**j**) the corresponding contour plot of Ru-hydroxide for $H_2O_2$ decomposition. In (**i**, **j**), a.u. indicates the arbitrary units. Source data are provided as a Source Data file.

oxide and hydroxide formulations. Gene set enrichment analysis (GSEA) is conducted on the gene signatures associated with various biological processes (Fig. 5i–k). The upregulation of 'Positive regulation of cell-substrate adhesion', 'Positive regulation of actin filament polymerization', and 'Focal adhesion' aligns with the observed

maintenance of focal adhesions and stem cell morphology facilitated by Ru-hydroxide. These results disclose the ability of Ru-hydroxide to maintain redox homeostasis, thus providing efficient protection and ROS damage defense on the stem cell living status from oxidative stress (Fig. 5l).

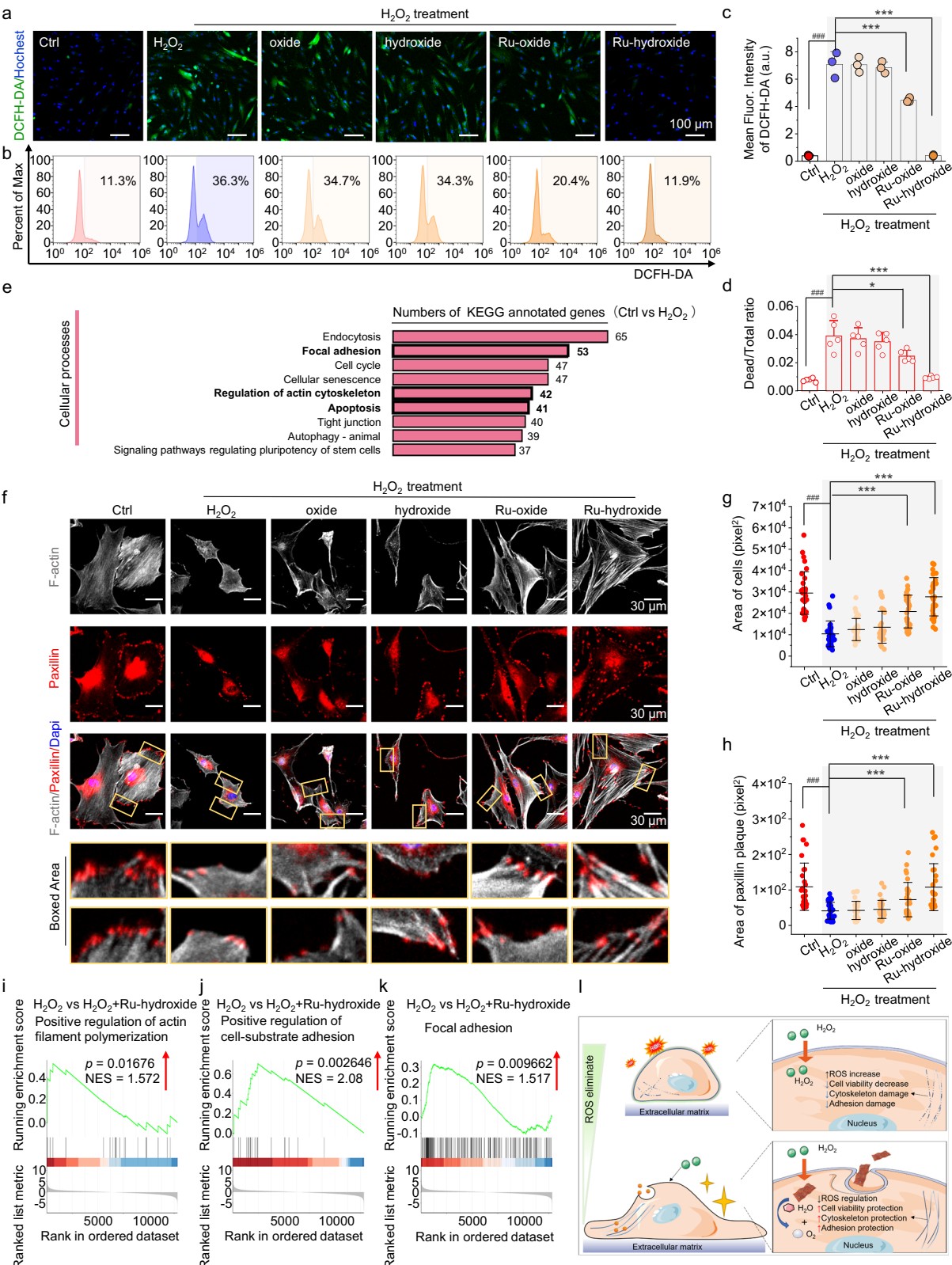

Then, we conduct a comprehensive investigation into the protective mechanisms of Ru-hydroxide on hMSCs, specifically examining its capacity to mitigate DNA damage and apoptosis in elevated ROS environments. Transcriptomic analysis is performed on three experimental groups: Ctrl (PBS), $H_2O_2$, and Ru-hydroxide ($H_2O_2$+Ru-hydroxide) treated hMSCs. The transcriptomic data demonstrate good biological replication as evidenced by unguided principal component

analysis (PCA; Fig. 6a). The top-ranking KEGG pathways between $H_2O_2$ and Ru-hydroxide treatments (Fig. 6b) suggest that Ru-hydroxide mediates oxidative stress protection of hMSCs in ROS-enriched microenvironments by the FoxO signaling pathway[68] and HIF-1 signaling pathway[69]. The beneficial effects of Ru-hydroxide on hMSCs are manifested across multiple cellular processes, including cytoskeletal organization and focal adhesion (regulation of actin cytoskeleton and

**Fig. 5 | In vitro ROS damage defense of stem cells via Ru-hydroxide.**
**a** Fluorescence images, Ctrl (hMSCs+PBS), (**b**) flow cytometry, and (**c**) mean fluorescence intensity of DCFH-DA staining ($n = 3$ independent replicates), $^{###}p_{(H2O2)} < 0.0001$, $^{***}p_{(Ru\text{-}oxide+H2O2)} < 0.0001$, $^{***}p_{(Ru\text{-}hydroxide+H2O2)} < 0.0001$. Scale bar: 100 μm. **d** The numbers of cells calculated from live/dead staining ($n = 5$ independent replicates), $^{###}p_{(H2O2)} < 0.0001$, $^{*}p_{(Ru\text{-}oxide+H2O2)} = 0.0179$, $^{***}p_{(Ru\text{-}hydroxide+H2O2)} < 0.0001$. **e** KEGG classification of cellular processes between Ctrl and H$_2$O$_2$. **f** Fluorescence images of F-actin and paxillin (Gray: F-actin, Red: paxillin, Blue: Dapi). Scale bar: 30 μm. **g** Quantitative analysis of the area of cells ($n = 30$ independent replicates), $^{###}p_{(H2O2)} < 0.0001$, $^{***}p_{(Ru\text{-}oxide+H2O2)} < 0.0001$, $^{***}p_{(Ru\text{-}hydroxide+H2O2)} < 0.0001$. **h** Quantitative analysis of area of focal adhesion plaque ($n = 30$ independent

replicates), $^{###}p_{(H2O2)} < 0.0001$, $^{***}p_{(Ru\text{-}oxide+H2O2)} < 0.0001$, $^{***}p_{(Ru\text{-}hydroxide+H2O2)} < 0.0001$. **i–k** GSEA analysis of H$_2$O$_2$ versus Ru-hydroxide+H$_2$O$_2$. **l** Schematic illustration of stem cell effects upon the addition of high-concentration H$_2$O$_2$ and ROS damage defense via Ru-hydroxide. In (**c–h**), data are presented as mean values ± SD, $^{*}p < 0.05$, $^{**}p < 0.01$, $^{***}p < 0.001$, $^{#}p < 0.05$, $^{##}p < 0.01$, $^{###}p < 0.001$, ns, no significance. In (**c–h**), statistical significance was calculated using one-way ANOVA followed by Tukey's *post-hoc* test for multiple comparisons, all tests were two-sided. In (**i–k**), *p*-value obtained from one-sided Permutation test without multiple comparisons, NES indicates the normalized enrichment score. Experiments were repeated independently (**a–f**, **i–k**) three times with similar results. In (**c**), a.u. indicates the arbitrary units. In (**a–d**, **f–h**), Ctrl indicates hMSCs+PBS. Source data are provided as a Source Data file.

focal adhesion[70]), DNA damage response mechanisms (p53 signaling pathway[71]), cellular homeostasis (ECM-receptor interaction, cell senescence, and cell cycle regulation), and cell apoptosis (apoptosis)[71]. Furthermore, Ru-hydroxide exhibited protective effects through modulation of inflammatory responses (via mTOR[72] and TNF signaling[73]), regulation of stem cell development and matrix formation (signaling pathways regulating pluripotency of stem cells, ECM-receptor interaction, and MAPK signaling pathwa[74]), and osteogenic differentiation (via Hippo[75], PI3K-Akt[76], and TGF-beta signaling pathways[77]). The top-ranking Gene Ontology (GO) term enrichment analysis speculates these resembling cellular processes (Fig. 6c). Thus, we then detect stem cell protection of Ru-hydroxide in aspects of cellular DNA damage, apoptosis, inflammation, and osteogenesis.

Moreover, GSEA is performed to uncover the gene signatures associated with various biological processes. (Fig. 6d,e). Compared to the H$_2$O$_2$ group, the Ru-hydroxide-treated group demonstrates downregulation of pathways involved in 'Intrinsic apoptotic signaling pathway' and 'Apotosis-multiple species', indicating its cytoprotective effects on stem cells through attenuation of oxidative stress-induced apoptosis. It has been recognized that excessive ROS levels can cause an irreversible attack on DNA[78]. Subsequently, we comprehensively investigate the therapeutic potential of Ru-hydroxide in ameliorating ROS-mediated DNA damage and cellular apoptosis. We first evaluate the phosphorylation status of histone H2A.X (γ-H2A.X), a well-characterized biomarker of double-strand DNA breaks. Immuno-fluorescence microscopy and nuclear spectral analysis demonstrate pronounced γ-H2A.X signal co-localizing with DAPI in the H$_2$O$_2$ group (Fig. 6f), whereas Ru-hydroxide treatment significantly attenuates this signal to levels comparable to Ctrl conditions, as quantitatively demonstrated in Fig. 6i. Additionally, as depicted in Fig. 6g and Supplementary Fig. 29, compared the Ctrl group, the DNA/RNA damage signals and the oxidative DNA damage (8-Oxogauanine) markers in nuclei are significantly elevated in the H$_2$O$_2$ group, while the Ru-hydroxide group display substantially reduced red signals, resembling the fluorescence levels in the Ctrl group (Fig. 6h,j). Furthermore, we have evaluated DNA damage responses using more relevant assays like alkaline and neutral comet assays for single and double-strand breaks (Supplementary Fig. 30). Compared to H$_2$O$_2$ treatment, comet assays reveal a significant decrease in DNA damage with Ru-hydroxide treatment. The p-ATM and p-ERK were known to participate in ROS-induced DNA damage response[79–81]. Thereafter, we perform an analysis of p-ATM and p-ERK by immunofluorescence staining. The experimental findings demonstrate that H$_2$O$_2$ exposure led to elevated protein expression levels, whereas specimens treated with Ru-hydroxide exhibited markedly diminished signaling patterns comparable to the Ctrl group, as validated through quantitative analyses (Supplementary Figs. 31, 32). Flow cytometry analysis partitions cell populations into stages of early and late apoptosis. (Fig. 6l,m and Supplementary Figs. 33, 34). While the H$_2$O$_2$-treated group displays a pronounced apoptotic percentage, the Ru-hydroxide-treated group maintains an apoptotic percentage analogous to the Ctrl group, suggesting effective cytoprotection against oxidative stress-mediated cell apoptosis

and preservation of cellular redox homeostasis comparable to baseline conditions (Fig. 6k).

## Bone tissue regeneration in inflammatory mandible defects
Inflammation is intimately related to oxidative stress and redox imbalance[82,83]; the in vivo bone tissue and endogenous stem cell microenvironment provided by Ru-hydroxide is evaluated in an inflammatory mandible defect model (Fig. 7a). Initially, we establish and validate a lipopolysaccharide (LPS)-induced ROS-associated mandible defect model (referred to as ROSup) through DHE (dihydroethidium) fluorescence detection (Supplementary Fig. 35). Our findings demonstrate a marked reduction in ROS levels following Ru-hydroxide administration. The Ru-hydroxide treatment exhibits favorable biocompatibility, with no observable adverse effects on viscus (heart, liver, spleen, lung, and kidney; Supplementary Fig. 36), hematological index (Supplementary Fig. 37), or body weight (Supplementary Fig. 38). Immunofluorescence analysis (Fig. 7b) and subsequent quantification (Fig. 7c,d) reveal that Ru-hydroxide significantly downregulated pro-inflammatory mediators (TNF-α and IL-1β), thereby revealing the inflammatory status within the ROS-associated mandible defect region.

Stem cells naturally expressing high levels of CD44 play a crucial role in facilitating adhesive interactions with the bone marrow vasculature, thereby promoting their homing to endosteal surfaces and contributing to the early-stage wound healing of natural bone as well as bone regeneration[84,85]. Therefore, we further investigate the protective effects of Ru-hydroxide on endogenous stem cells in vivo for 1 week by histology staining (Fig. 7e)[86]. Confocal scanning images of CD44 and γ-H2A.X dual staining and the linear distribution results clearly indicate an increased migration and adhesion of CD44$^+$ cells to the defect region in the Ru-hydroxide-ROSup group, as opposed to the ROSup group (Fig. 7f). In addition, γ-H2A.X expression is the highest in the ROSup group, which decreases obviously in the Ru-hydroxide-ROSup group (Fig. 7e, f). The quantitative analysis of the γ-H2A.X fluorescence intensity is shown in Fig. 7g. These phenomena indicate that Ru-hydroxide can provide a more favorable microenvironment in ROS-associated bone defects for stem cells, as evidenced by reducing cellular DNA damage and increasing endogenous stem cell recruitment (Fig. 7h).

After validating that Ru-hydroxide can effectively inhibit endogenous inflammation and restore redox homeostasis in mandible defect, we subsequently conduct a comprehensive evaluation of its efficacy in enhancing bone tissue regeneration within inflammatory microenvironments. First, we utilize the osteogenic differentiation of hMSCs in excessive ROS conditions in vitro to disclose the application potential of Ru-hydroxide. The group treated with H$_2$O$_2$ exhibits a reduced expression of alkaline phosphatase (ALP) and diminished formation of calcium nodules (assessed by Alizarin Red staining), as displayed in Fig. 8a. Nevertheless, the osteogenic phenotype weakened by H$_2$O$_2$ can be restored with the introduction of Ru-hydroxide. The quantitative analysis of the ALP and Alizarin Red staining area is shown in Fig. 8b, c, suggesting that the high level of

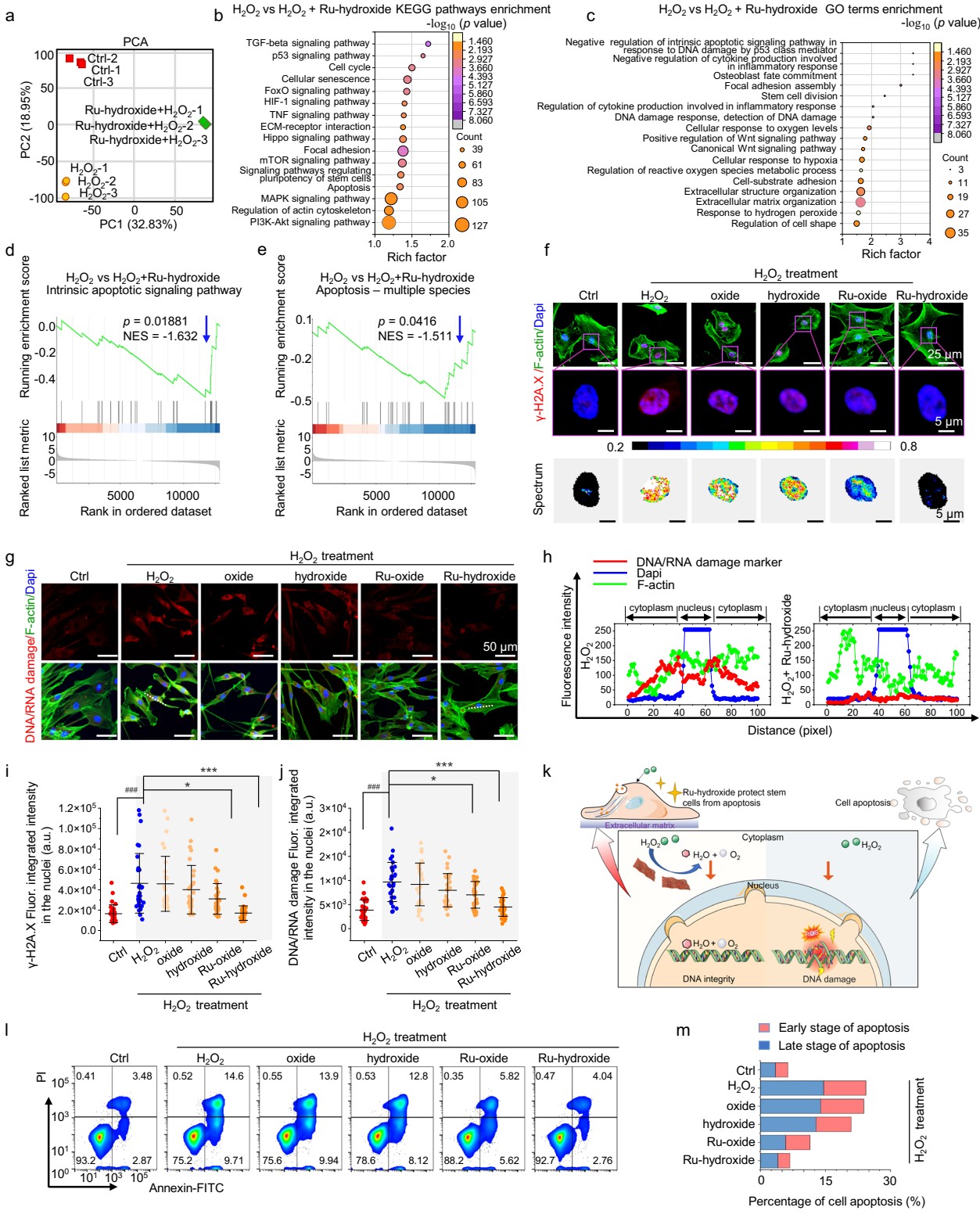

$H_2O_2$ can't block the hMSC osteogenesis with the supplementation of Ru-hydroxide.

It was reported that the early bone formation marker ALP presented a quick positive regulation, with the ALP detected at higher levels during the process of bone regeneration[87]. Additionally, BMP2/4 has been recognized as a key growth factor for osteogenesis and osseous defect repair[88]. Thereafter, by utilizing an inflammatory mandible defect model, we examine the expression of ALP matrix and BMP2/4 secretion during the early endogenous stem cell

differentiation stage, while mineralization at later stages is evaluated using micro-CT imaging. Immunofluorescence staining of osteoinductive markers reveals the enhanced expression levels in the Ru-hydroxide-ROSup group (Fig. 8d). Quantitative analysis further confirmed the highest expression of BMP2/4 and ALP in the Ru-hydroxide-ROSup group (Fig. 8e, f), reflecting its potential for promoting osteogenesis. Briefly, the results suggest that Ru-hydroxide can function as a biocatalytic ROS-scavenging material to efficiently protect endogenous bone-related stem cells from

**Fig. 6 | hMSCs protection mechanisms of Ru-hydroxide on reversing ROS attack-related DNA damage and apoptosis. a** PCA analysis between samples. **b** Enriched KEGG pathway involved DNA damage between $H_2O_2$ and Ru-hydroxide $+H_2O_2$. **c** Enriched GO terms of $H_2O_2$ versus Ru-hydroxide+$H_2O_2$. **d, e** GSEA analysis of intrinsic apoptotic signaling pathway and apoptosis between $H_2O_2$ and Ru-hydroxide+$H_2O_2$. **f** Fluorescence images and spectrum images of γ-H2A.X. **g** Fluorescence images of DNA/RNA damage. Scale bar: 50 μm. **h** Linear distribution of fluorescence intensity from DNA/RNA damage staining. **i** Quantitative analysis of fluorescence intensity from γ-H2A.X staining in the nuclei ($n = 30$ independent replicates), $^{###}p_{(H2O2)} < 0.0001$, $^{*}p_{(Ru\text{-}oxide+H2O2)} = 0.0491$, $^{***}p_{(Ru\text{-}hydroxide+H2O2)} < 0.0001$. **j** Quantitative analysis of fluorescence intensity from DNA/RNA damage staining in the nuclei ($n = 30$ independent replicates), $^{###}p_{(H2O2)} < 0.0001$, $^{*}p_{(Ru\text{-}oxide+H2O2)} = 0.0225$, $^{***}p_{(Ru\text{-}hydroxide+H2O2)} < 0.0001$. **k** Schematic illustration of DNA damage upon the treatment of high-concentration $H_2O_2$ and Ru-hydroxide protection. **l** Apoptosis analysis was performed using flow cytometry of Annexin V-FITC/PI stained hMSCs. **m** The cell percentages in stages of early apoptosis and late apoptosis. In (**i, j**), data are presented as mean values ± SD, $^{*}p < 0.05$, $^{**}p < 0.01$, $^{***}p < 0.001$, $^{#}p < 0.05$, $^{##}p < 0.01$, $^{###}p < 0.001$, ns, no significance. In (**b, c**), $p$-value obtained from one-sided Hypergeometric test without multiple comparisons. In (**d, e**), $p$-value obtained from one-sided Permutation test without multiple comparisons, NES indicates the normalized enrichment score. In (**i, j**), statistical significance was calculated using one-way ANOVA followed by Tukey's post-hoc test for multiple comparisons, all tests were two-sided. Experiments were repeated independently (**a–l, m**) three times with similar results. In (**i, j**), a.u. indicates the arbitrary units. In (**a–m**), Ctrl indicates hMSCs+PBS. Source data are provided as a Source Data file.

oxidative stress while preserving their cellular functions associated with osteogenesis (Fig. 8g).

Thereafter, micro-CT image analysis and 3D reconstruction reveal enhanced mineralized osseous matrix deposition at 2 weeks post-intervention, progressing to a complete bone structure at 8 weeks with the treatment of Ru-hydroxide (Fig. 8h). In contrast, both ROSup and hydroxide-ROSup group exhibit significant osseous defects, while the Ctrl group demonstrates limited new bone formation at the margins of defect at 8 weeks (Fig. 8h). Quantitative osteomorphometric parameters, including bone volume/tissue fraction (BV/TV), bone mineral density (BMD), trabecular thickness (Tb.Th), and trabecular number (Tb.N), are assessed and presented in Fig. 8i–l. The Ru-hydroxide-ROSup treatment yielded the highest BV/TV ratios compared to all other experimental groups (Fig. 8i). This trend is also observed in the analysis of BMD (Fig. 8j). Furthermore, the values of Tb.N and Tb.Th of Ru-hydroxide-ROSup is significantly higher than that of the ROSup group (Fig. 8k, l). The increased parameters along with the comprehensive 3D reconstruction images of the Ru-hydroxide-ROSup group indicate a denser structure of the newly formed bone. Additionally, the observation that Ru-hydroxide undergoes degradation during the treatment cycle indicates its efficient impact on the early stage of bone reconstruction (Supplementary Fig. 39).

## Discussion

In summary, by mimicking the natural IADS mechanism for efficient redox reaction, we have proposed the de novo bioinspired design of an efficient artificial antioxidase using Ru-hydroxide for superior redox homeostasis and maxillofacial bone regeneration. Our studies validate that the hydroxyl-synergistic Ru catalytic sites can react with oxygen species efficiently and complete rapid proton and electron transfer, thus significantly reducing the activation energy of antioxidative reactions. Accordingly, the Ru-hydroxide exhibits superior and simultaneous SOD-, CAT-, and GPx-mimetic activities for ROS elimination, as well as favorable cycling stability for long-term usage. Consequently, Ru-hydroxide demonstrates the antioxidase-like ability to sustain stem cell viability and osteogenic differentiation in elevated ROS environments by preventing oxidative stress-induced DNA damage and cell apoptosis, thereby supporting metabolic processes, developmental pathways, osteogenesis functions, and mitigating inflammatory responses. Strikingly, the synthesized Ru-hydroxide exhibits efficient in vivo biocatalytic microenvironment modulations and repair functions during inflammatory maxillofacial defect regeneration.

Our proof-of-concept design opens up a way to construct efficient and stable biocatalytic materials that are inspired by natural IADS mechanisms for maintaining superior redox homeostasis. Our results also indicate that the innovative hydroxyl-synergistic monoatomic Ru centers in Ru-hydroxide enable rapid and sustained proton/electron transfer for effective ROS biocatalysis. Furthermore, the high electron density of the Ru centers optimizes the binding strength of oxygen species and enhances resistance to $H_2O_2$ toxicity in Ru-hydroxide. Therefore, Ru-hydroxide exhibits superior and broad-spectrum antioxidase-like activities for ROS elimination, as well as favorable cycling stability for long-term use. This bioinspired strategy that can mimic the proton/electron transfer properties of natural enzymes to overcome the cellular redox imbalance may offer a hopeful route to develop antioxidase-like biocatalysts for treating a broad range of inflammation-associated diseases, such as arthritis, diabetic wounds, enteritis, and bone fractures.

This investigation centers on modulating the microenvironments of endogenous stem cells through biocatalytic materials for the regeneration of inflammatory mandibular tissues. Following injuries, the mobilized endogenous stem cells migrate toward the injury site and actively engage in tissue regeneration. This work demonstrates that the Ru-hydroxide with antioxidase-like abilities can sustain stem cell viability and osteogenic differentiation in elevated ROS conditions by preventing oxidative stress-induced DNA damage and cell apoptosis, which may meet the clinical demands in maxillofacial bone regeneration. Moreover, the in situ rapid expansion of endogenous stem cells shows promise for effectively regenerating complex maxillofacial bone defects and presents facile application potential to be integrated with many other therapeutic strategies.

The future direction of our study involves exploring the detailed long-term biocompatibility and expanding the clinical applicability and therapeutic scopes. In the current study, a single model of maxillofacial bone defect was utilized to assess the stem cell-based therapeutic impacts of Ru-hydroxide. Our vision includes broadening this investigation to encompass other models of oxidative stress-related diseases and aligning them with established clinical protocols, as well as conducting comprehensive biological studies to accelerate their clinical translation. Furthermore, deeper therapeutic mechanisms on stem cells and tissues should be evaluated, such as the long-term biological effects on gene and immunity homeostasis. We anticipate that the proposed strategy and reaction mechanism in this work could be extended to many other biocatalytic and bioinspired materials, such as ROS-generating materials, cascade artificial enzyme systems, and Janus biocatalysts. Additionally, by combining this strategy with immune-regulating factors or functional cells, biomedical and therapeutic avenues for clinical utilization can be obtained.

## Methods

### Ethical statement

Animal experiments and procedures, including euthanasia, were conducted following protocols approved by the Institutional Animal Care and Use Committee at Sichuan University (Number: WCHSIRB-D-2020-361). Additionally, the Laboratory Animal Welfare and Ethics Committee of West China Hospital of Stomatology reviewed and approved the study. All experiments involving animal use were performed in accordance with the ARRIVE guidelines.

### Materials

The ruthenium (III) chloride hydrate ($RuCl_3 \cdot xH_2O$) was sourced from Energy Chemical (Anhui, China). The hexamethylenetetramine (HMT,

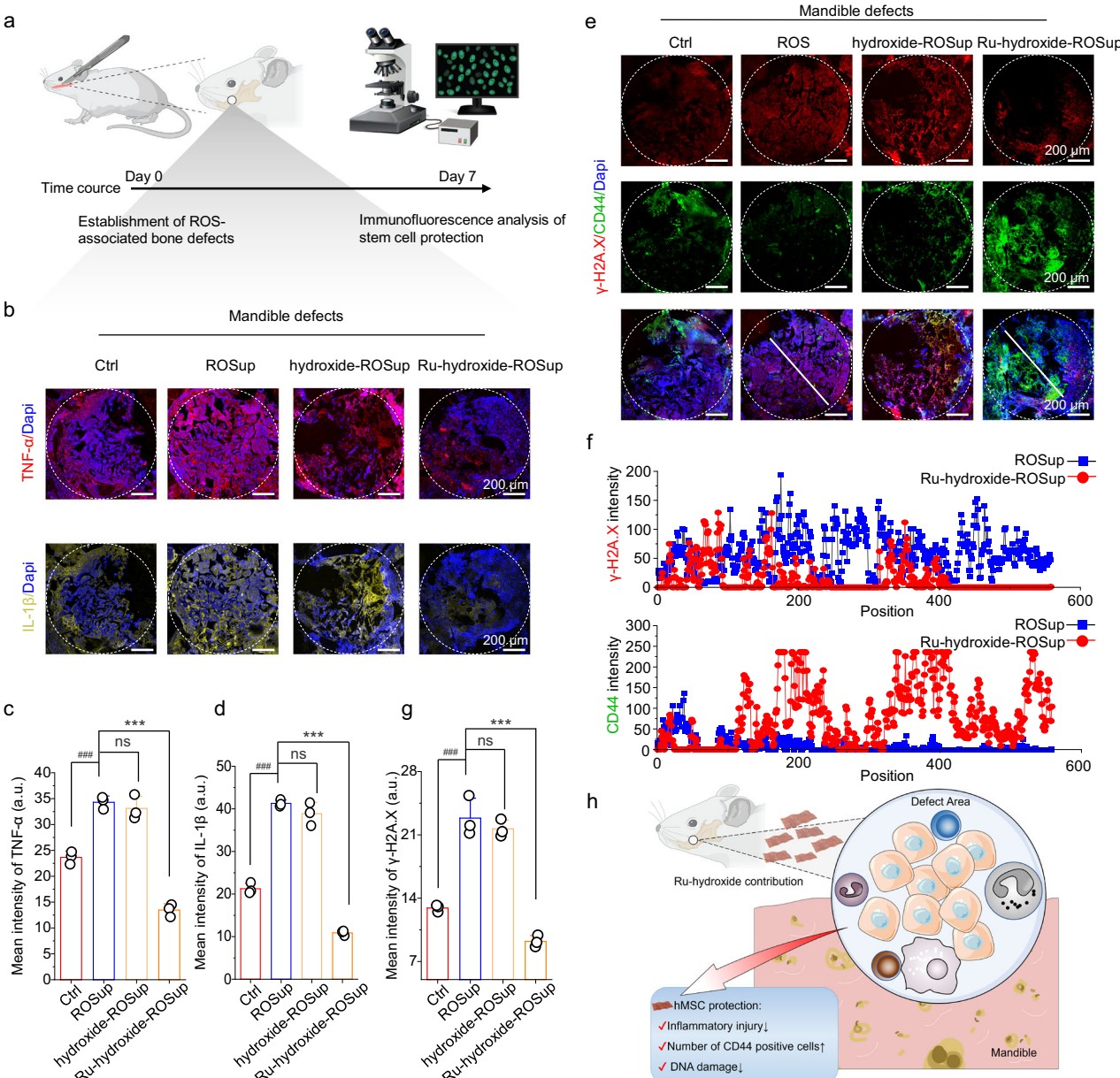

**Fig. 7 | Endogenous inflammation inhibition in LPS-induced ROS-associated mandible defect by Ru-hydroxide. a** Schematic illustration of animal model of ROS-associated bone defects. **b** Fluorescence images of TNF-α and IL−1β at week 1 after operation. **c** Quantitative analysis of fluorescence mean intensity of TNF-α ($n = 3$ independent replicates), $^{\#\#\#}p_{(ROSup)} = 0.0002$, $p_{(hydroxide-ROSup)} = 0.7954$, $^{***}p_{(Ru-hydroxide-ROSup)} < 0.0001$. **d** Mean intensity of IL−1β ($n = 3$ independent replicates), $^{\#\#\#}p_{(ROSup)} < 0.0001$, $p_{(hydroxide-ROSup)} = 0.2970$, $^{***}p_{(Ru-hydroxide-ROSup)} < 0.0001$. **e** Fluorescence images of γ-H2A.X. **f** Linear fluorescence distribution of γ-H2A.X. **g** Mean intensity of γ-H2A.X ($n = 3$ independent replicates), $^{\#\#\#}p_{(ROSup)} < 0.0001$,

$p_{(hydroxide-ROSup)} = 0.6664$, $^{***}p_{(Ru-hydroxide-ROSup)} < 0.0001$. **h** Schematic illustration of bone defects. In (**c**–**g**), data are presented as mean values ± SD, $^{*}p < 0.05$, $^{**}p < 0.01$, $^{***}p < 0.001$, $^{\#}p < 0.05$, $^{\#\#}p < 0.01$, $^{\#\#\#}p < 0.001$, ns, no significance; statistical significance was calculated using one-way ANOVA followed by Tukey's post-hoc test for multiple comparisons, all tests were two-sided. Experiments were repeated independently (**b**–**f**) three times with similar results. In (**c**–**g**), a.u. indicates the arbitrary units. In (**b**–**e**, **g**), Ctrl indicates mandible defects+saline. Source data are provided as a Source Data file.

---

99.0%), nickel chloride hexahydrate ($NiCl_2 \cdot 6H_2O$), and cobalt chloride hexahydrate ($CoCl_2 \cdot 6H_2O$) were sourced from Aladdin Reagents (Shanghai, China). All aqueous solutions were prepared using deionized water. Any additional reagents not explicitly mentioned were provided by Aladdin Reagents and all materials were of analytical grade, used as received.

### Characterization

X-ray diffraction (XRD) was conducted using a DX-2700BH (HaoYuan Instrument, China) with Cu Kα radiation over a 2θ range of 5–80°. Field emission SEM was conducted using the Hitachi Regulus8220

from Japan. For TEM, AC HAADF-STEM, and EDS mapping, a JEM ARM 200 F (Japan) operated at 200 kV was utilized. XPS measurements were carried out on the K-Alpha™ + X-ray Photoelectron Spectrometer System (Thermo Scientific), which features a Hemispheric 180° dual-focus analyzer with a 128-channel detector. The X-ray absorption (XAS) spectra of Ru K-edge were obtained in fluorescence mode at the BL14W1 beamline of the Shanghai Synchrotron Radiation Facility, China, operated at 3.5 GeV with maximum injection currents of 230 mA. The structure analysis of materials was measured by Raman spectroscopy (XploRA PLUS, HORIBA). The cyclic voltammetry (CV) curves were conducted via Gamry reference 600

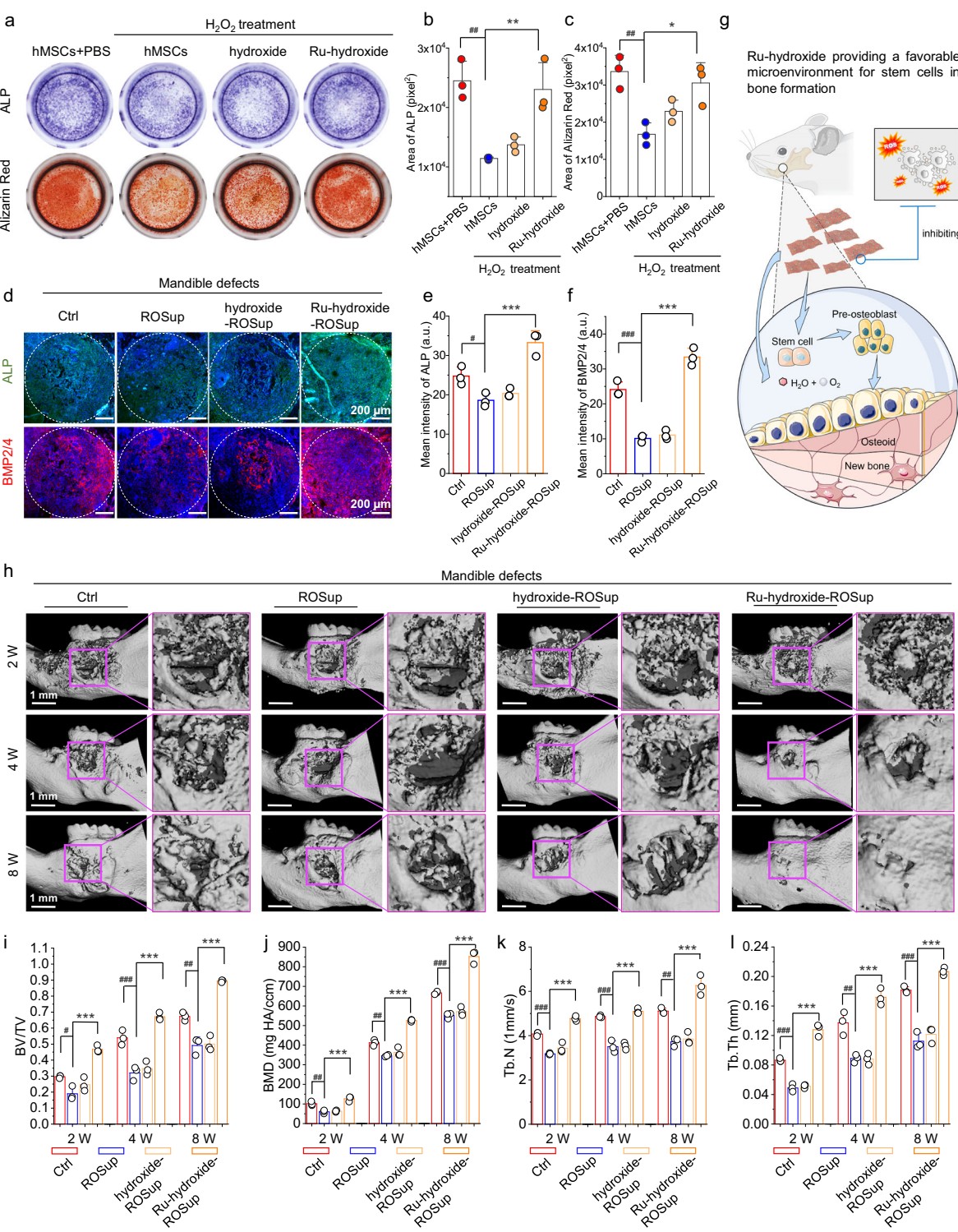

workstations (Gamry, USA). In-situ FTIR measurements were conducted using an infrared spectrometer (Thermo Scientific, iS50 FTIR) equipped with an in-situ spectrum cell (Shanghai Yuanfang Technology Co., Ltd., SPECEL-III). The absorbance was measured using a multifunctional enzyme labeler (ReadMax 1900). Data analysis was performed with various software, including MDI Jade 6, Digital Micrograph 3.7.4, Avantage 5.967, Artemis software 0.9.26, Athena software 0.9.26, Adobe Illustrator 27.0.1, VASP 5.4.1, Origin 2022, GraphPad Prism 8.0, Image-Pro® Plus 6.0, Image J 1.53c, and FlowJo v10.8.1. Bioinformatics analyses were performed on the free online platform BMKCloud (www.biocloud.net).

## Synthesis of antioxidase-like Ru-hydroxide

The synthesis commenced with the preparation of two precursor solutions: Solution A, comprising $NiCl_2 \cdot 6H_2O$ (149 mg) and $CoCl_2 \cdot 6H_2O$ (297.5 mg) in deionized water (10 mL), and Solution B, containing HMT (3.154 g) in deionized water (25 mL). The solutions were combined under continuous agitation at ambient temperature, followed by hydrothermal treatment at 95 °C for 5 h in a Teflon-lined autoclave. The resultant CoNi LDH biocatalytic material (named hydroxide) was isolated via centrifugation (9558 × $g$, 5 min), which was purified via sequential washing with deionized water and ethanol before vacuum drying at 60 °C overnight. The synthesis of Ru-

**Fig. 8 | Bone regeneration in mandibular defects by Ru-hydroxide. a** ALP staining after 3-day in vitro osteo-induction and Alizarin Red staining after 21-day in vitro osteo-induction using hMSCs. Quantitative analysis of (**b**) ALP ($n = 3$ independent replicates), $^{\#\#}p_{(hMSCs+H2O2)} = 0.0023$, $^{**}p_{(Ru\text{-}hydroxide+H2O2)} = 0.0049$, and (**c**) Alizarin Red ($n = 3$ independent replicates) in $H_2O_2$ treated MSCs, $^{\#\#}p_{(hMSCs+H2O2)} = 0.0044$, $^{*}p_{(Ru\text{-}hydroxide+H2O2)} = 0.0142$. **d** Fluorescence images of ALP and BMP2/4 at week 2 after the mandible defects operation. Quantitative analysis of (**e**) ALP ($n = 3$ independent replicates), $^{\#}p_{(ROSup)} = 0.0341$, $^{***}p_{(Ru\text{-}hydroxide\text{-}ROSup)} = 0.0002$, and (**f**) BMP2/4 ($n = 3$ independent replicates) in the inflammatory mandible defects, $^{\#\#\#}p_{(ROSup)} < 0.0001$, $^{***}p_{(Ru\text{-}hydroxide\text{-}ROSup)} < 0.0001$. **g** Schematic illustration of Ru-hydroxide contributing to bone formation. **h** 3D reconstruction of micro-CT images. New bone formation quantitative analysis of (**i**) BV/TV ($n = 3$ independent replicates), $^{\#}p_{(ROSup, 2W)} = 0.0168$, $^{***}p_{(Ru\text{-}hydroxide\text{-}ROSup, 2W)} < 0.0001$, $^{\#\#\#}p_{(ROSup, 4W)} = 0.0007$, $^{***}p_{(Ru\text{-}hydroxide\text{-}ROSup, 4W)} < 0.0001$, $^{\#\#}p_{(ROSup, 8W)} = 0.0017$, $^{***}p_{(Ru\text{-}hydroxide\text{-}ROSup, 8W)} < 0.0001$. **j** BMD ($n = 3$

independent replicates), $^{\#\#}p_{(ROSup, 2W)} = 0.0031$, $^{***}p_{(Ru\text{-}hydroxide\text{-}ROSup, 2W)} = 0.0001$, $^{\#\#}p_{(ROSup, 4W)} = 0.0019$, $^{***}p_{(Ru\text{-}hydroxide\text{-}ROSup, 4W)} < 0.0001$, $^{\#\#\#}p_{(ROSup, 8W)} = 0.0009$, $^{***}p_{(Ru\text{-}hydroxide\text{-}ROSup, 8W)} < 0.0001$. **k** Tb.N ($n = 3$ independent replicates), $^{\#\#\#}p_{(ROSup, 2W)} = 0.0002$, $^{***}p_{(Ru\text{-}hydroxide\text{-}ROSup, 2W)} < 0.0001$, $^{\#\#\#}p_{(ROSup, 4W)} < 0.0001$, $^{***}p_{(Ru\text{-}hydroxide\text{-}ROSup, 4W)} < 0.0001$, $^{\#\#}p_{(ROSup, 8W)} = 0.0017$, $^{***}p_{(Ru\text{-}hydroxide\text{-}ROSup, 8W)} < 0.0001$, and (**l**) Tb.Th ($n = 3$ independent replicates), $^{\#\#\#}p_{(ROSup, 2W)} < 0.0001$, $^{***}p_{(Ru\text{-}hydroxide\text{-}ROSup, 2W)} < 0.0001$, $^{\#\#}p_{(ROSup, 4W)} = 0.0025$, $^{***}p_{(Ru\text{-}hydroxide\text{-}ROSup, 4W)} < 0.0001$, $^{\#\#\#}p_{(ROSup, 8W)} < 0.0001$, $^{***}p_{(Ru\text{-}hydroxide\text{-}ROSup, 8W)} < 0.0001$. In (**b–l**), data are presented as mean values ± SD, $^{*}p < 0.05$, $^{**}p < 0.01$, $^{***}p < 0.001$, $^{\#}p < 0.05$, $^{\#\#}p < 0.01$, $^{\#\#\#}p < 0.001$, ns, no significance; statistical significance was calculated using one-way ANOVA followed by Tukey's post-hoc test for multiple comparisons, all tests were two-sided. Experiments were repeated independently (**a–h**) three times with similar results. In (**e**, **f**), a.u. indicates the arbitrary units. In (**d–l**), Ctrl indicates mandible defects+saline. Source data are provided as a Source Data file.

---

hydroxide was achieved by dispersing and stirring 50 mg of the as-prepared hydroxide in 50 mL aqueous $RuCl_3 \cdot xH_2O$ (10 mg) solution under ambient conditions for 24 h. The resultant gray precipitate was isolated via centrifugation ($9558 \times g$, 5 min), followed by triple washing with water and ethanol and vacuum drying at 60 °C overnight. To investigate the effect of Ru loading, analogous samples were prepared using varying concentrations of $RuCl_3 \cdot H_2O$ precursor (2.5 mg and 25 mg).

## Synthesis of antioxidase-like Ru-oxide

The synthesized hydroxide precursor was subjected to thermal dehydrogenation under oxidative conditions (400 °C, 3 h) to generate a dehydrogenated CoNi oxide substrate. Subsequently, an aqueous solution was prepared by dissolving $RuCl_3 \cdot xH_2O$ (10 mg) in deionized water (50 mL). The oxide substrate (50 mg) was introduced into this solution, followed by continuous agitation at 50 °C for 24 h. The resulting gray precipitate was isolated via high-speed centrifugation ($9558 \times g$, 5 min) and underwent sequential purification with water and ethanol three times. The purified product was subjected to vacuum desiccation overnight at 60 °C, yielding the final Ru-oxide composite material.

## Details for in-situ FTIR measurement

In-situ FTIR spectroscopic analysis was performed using an infrared spectrometer (Thermo Scientific, iS50 FTIR) equipped with an in-situ spectrum cell (Shanghai Yuanfang Technology Co., Ltd., SPE-CEL-III). We typically prepare a Nafion solution consisting of 210 μL of isopropyl alcohol, 750 μL of deionized water, and 40 μL of Nafion (Perfluorosulfonic acid ion exchange resin, Energy Chemical, 5% w/w in 1-propanol and water). Subsequently, a catalyst/Nafion dispersion was formulated to achieve a final concentration of 10 mg/mL. During the experiment, we deposited 40 μL of the catalyst solution onto a ZnSe crystal surface, allowed it to dry, and then installed the crystal into the in-situ cell. Subsequently, 5 mL of 0.5 M $H_2O_2$ was included in the in-situ cell, and we collected the in-situ FTIR spectra at specific intervals, maintaining a reaction time of 10 min.

## SOD-like catalytic activity test

To catalytically eliminate $\cdot O_2^-$ radicals, potassium superoxide ($KO_2$, 1 mg) was solubilized in dimethyl sulfoxide (DMSO) containing 18-crown-6-ether (3 mg/mL). Subsequently, Ru-hydroxide was added to this $KO_2$/DMSO solution to achieve a 50 μg/mL concentration. Following a 5 min incubation period, residual $\cdot O_2^-$ levels were quantified using nitroblue tetrazolium as a chromogenic probe (10 μL of 10 mg/mL nitroblue tetrazolium in DMSO). The scavenging efficiency was determined by measuring the absorption intensity at $\lambda_{max} = 680$ nm and comparing it to the initial $\cdot O_2^-$ concentration.

## CAT-like catalytic activity test

The CAT-like activity was evaluated through a dual assessment of $H_2O_2$ decomposition and molecular $O_2$ evolution. The $H_2O_2$ scavenging efficiency was quantified by preparing a reaction mixture containing the biocatalyst (50 μg/mL) and $H_2O_2$ (10 mM) in PBS buffer (pH 7.4). The solution (50 μL) was collected at predetermined time intervals (3, 6, 9, and 12 min) and combined with 100 μL of titanium (IV) sulfate solution (13.9 mM) for spectrophotometric analysis at 405 nm to determine residual $H_2O_2$ concentrations.

To assess $O_2$ generation capability, the reaction system was established by introducing $H_2O_2$ (10 M, 200 μL) into a PBS buffer solution (pH 7.4, 20 mL) containing Ru-hydroxide catalyst (10 mg/mL, 12 μL). Real-time monitoring of dissolved oxygen concentration was performed using a Dissolved Oxygen Meter, with measurements recorded at 5-s intervals over a 3-min reaction period.

The CAT-like activity kinetics were evaluated through substrate-dependent analysis by modulating $H_2O_2$ concentrations. The reaction mixture comprised 20 mL PBS (pH 7.4) and 12 μL of Ru-hydroxide catalyst (10 mg/mL) in a 50 mL centrifuge vessel. $H_2O_2$ concentrations ranging from 5 to 400 mM were introduced to the reaction system, and $O_2$ solubility variations were monitored over a 3 min period. Initial reaction velocities ($V_0$) were derived from temporal absorbance profiles for each substrate concentration. The reaction kinetics were analyzed by fitting the velocity ($V_0$)-substrate concentration (S) relationships to the Michaelis-Menten equation (Eq. (1)). The kinetic parameters, including maximum velocity ($V_{max}$) and Michaelis constant ($K_m$), were determined through Lineweaver-Burk double-reciprocal plot analysis (Eq. (2)). The catalytic efficiency was quantified as turnover number (TON), calculated using Eq. (3), with [$E_0$] denoting the concentration of catalytic centers in the Ru-hydroxide.

$$V_0 = (V_{max} \times [S])/(K_m + [S]) \tag{1}$$

$$1/V_0 = K_m/(V_{max} \times [S]) + (1/V_{max}) \tag{2}$$

$$\mathrm{TON} = V_{max}/[E_0] \tag{3}$$

## GPx-like catalytic activity test

A reaction mixture was formulated comprising β-NADPH (reduced coenzyme II tetrasodium salt, 33 μL, 10 mg/mL, Aladdin), GSH (glutathione, 62 μL, 10 mg/mL, Aladdin), GR (glutathione reductase, 100 μL, 17 U/mL, Sigma-Aldrich), $H_2O_2$ (25 μL, 0.01 M), and the biocatalyst (50 μg/mL) in pH 7.4 PBS (775 μL). GPx-like catalytic activity was assessed spectrophotometrically by monitoring the oxidation of NADPH at $\lambda = 340$ nm. For kinetic analysis, absorbance at 340 nm was recorded using a multifunctional enzyme labeling instrument in kinetic mode over a period of 5 min, with measurements taken at 6-s

intervals. The relationship between absorbance values (A) and NADPH concentration ([C]) was calculated using Eq. (4). Additionally, the consumption rate of NADPH was determined using Eq. (5).

$$[C] = 0.4688 \times A - 0.0963 \tag{4}$$

$$Rate = ([C_t] - [C_0])/t \tag{5}$$

### DPPH• scavenging activity

In the experimental protocol, a reaction mixture was prepared comprising 2 mL of ethanol solution containing DPPH• (2,2-diphenyl-1-picrylhydrazyl, sourced from Aladdin, Shanghai, China) and the biocatalyst, both at concentrations of 50 µg/mL. The mixture underwent a 30-min dark incubation period to facilitate the reaction, followed by spectrophotometric analysis at 519 nm.

### Theoretical calculations

The computational methodology employed DFT calculations using the Vienna ab initio simulation package (VASP)[89]. The electronic structure calculations incorporated spin-polarized projector augmented wave (PAW) pseudopotentials for core electron representation[90], while the generalized gradient approximation with Perdew-Burke-Ernzerhof functional (GGA-PBE) was utilized for electron exchange-correlation effects[91]. The calculations employed a plane-wave basis set with a 450 eV energy cutoff. Electronic convergence criteria were set to $10^{-5}$ eV for energy and 0.02 eV/Å for forces, with calculations performed at the Gamma point. The DFT-D3 method was implemented to account for Van der Waals interactions[92]. The computational model incorporated a slab configuration with a 15 Å vacuum layer along the z-axis to eliminate periodic boundary effects.

To quantitatively assess the binding ability of the loading materials, binding strength is defined by Eq. (6):

$$E_{ads} = E_{adsorbate/substrate} - E_{adsorbate} - E_{substrate} \tag{6}$$

Where $E_{adsorbate}$, $E_{subatrate}$, and $E_{adsorbate/subatrate}$ represent the gas-phase molecule, the clean substrate, and the total energy of the substrate with adsorbed species, respectively.

To investigate catalytic effects, the change in Gibbs free energy (ΔG) is calculated according to Eq. (7):

$$\Delta G = \Delta E + \Delta ZPE + \Delta H_{0 \rightarrow 298K} - T\Delta S \tag{7}$$

ΔE represents the differential energy value computed through DFT calculations, whereas ΔZPE, ΔH, and ΔS correspond to the variations in zero-point energy, enthalpy, and entropy, respectively, associated with the reaction kinetics. The thermodynamic parameters, specifically enthalpy and entropy values for ideal gas molecules, were referenced from standardized thermodynamic databases. Computational analyses were facilitated using the VASPKIT package[93], while Hirshfeld charge population analyses were executed utilizing the Multiwfn computational program[94].

### Cellular experiments and detection of intracellular ROS

hMSCs were obtained from Cyagen Biosciences (HUXMA-01001, Cyagen, China) and expanded in human mesenchymal stem cell growth medium (HUXMX-90021, Cyagen, China). The cells were cultured under standard conditions (37 °C, 5% CO$_2$) in accordance with the manufacturer's protocol. To ensure experimental consistency, cells from passages 4–6 were utilized at equivalent seeding densities across all experimental conditions, initiating treatments after overnight culture.

For the cellular protection assays, 100 µg/mL of synthetic biocatalysts and 100 µM of hydrogen peroxide were introduced sequentially. After a 1 h incubation, the medium was replaced with a standard culture medium (human mesenchymal stem cell growth medium, HUXMX-90021, Cyagen, China) and maintained until the designated testing time. For osteogenic differentiation studies, the standard medium was substituted with an osteogenic induction medium, and cells were incubated overnight.

To assess residual reactive oxygen species (ROS) levels, DCFH-DA (S0033S, Beyotime, China) was employed, and the procedure was conducted according to the manufacturer's guidelines. The intracellular ROS levels were quantified via flow cytometric analysis (AttuneTM NxT, Invitrogen, USA) and visualized through confocal laser scanning microscopy (FV3000, Olympus, Tokyo, Japan). In the flow cytometry procedure, cells were digested using 0.25% trypsin solution (EDTA-free), followed by centrifugation at 300 $\times g$ for 5 min and subsequent supernatant removal. After washing with PBS, the cells were incubated with a DCFH-DA working solution for 20 min. Following PBS washing, the cells were resuspended in PBS for analysis. Through gating strategies, small-sized cells/particles were excluded, as illustrated in Supplementary Fig. 24. The resulting figures were generated using Flowjo software (version 10.8.1).

### Live/dead staining

Cell viability assessment was performed utilizing dual fluorescence staining with Calcein AM and Propidium Iodide (PI) in accordance with the manufacturer's specifications (Beyotime, China). Specifically, live cells were labeled with 2 µM Calcein AM solution prepared in PBS (pH 7.4), while dead cells were identified using 4.5 µM PI solution. The stained cells were subsequently analyzed and quantified using a Celigo Image Cytometer (Nexcelom Bioscience LLC, USA).

### Immunofluorescence staining

Following H$_2$O$_2$ treatment or removal, cellular samples were maintained in culture until the testing time. The samples underwent two washes with PBS and then fixed in 4% paraformaldehyde solution (Biosharp, China) at room temperature for 10 min. Cell membrane permeabilization was achieved using 0.1% Triton X-100 in PBS for 10 min, followed by immunological blocking with 1% BSA in PBS for 60 min. Primary antibody solutions, prepared in 1% BSA-PBS, were applied to the samples and incubated at 4 °C overnight. Subsequently, fluorophore-conjugated secondary antibodies were introduced and incubated at room temperature under dark conditions. Nuclear counterstaining was performed using DAPI (10 µg/mL, Solarbio, China), and the samples were mounted using an antifading mounting medium (Solarbio, China). Triple PBS washing steps were implemented between each procedural stage. Microscopic analysis was conducted using an FV3000 confocal laser scanning microscope (Olympus, Tokyo, Japan).

The primary antibodies utilized in this study included DNA/RNA Damage (1:200 dilution, ab62623, Abcam, America), p-ATM (1:50 dilution, R380751, ZEN-BIOSCIENCE, China), p-ERK (1:50 dilution, R380698, ZEN-BIOSCIENCE, China), 8-oxoguanine (1:50 dilution, 130914, Santa Cruz Biotechnology, America), γH2A.X (phospho S139) antibody (1:200 dilution, ab81299, Abcam, America), Paxillin antibody (1:200 dilution, ab32084, Abcam, America), The cytoskeleton was stained with FITC-conjugated phalloidin (1:200 dilution, A12379, Invitrogen, America). The secondary antibodies included Cy3 goat anti-rabbit IgG (1:50 dilution, GB21303, Servicebio, China), Alexa Fluor 488 goat anti-rabbit IgG (1:200 dilution, ab150077, Abcam, America), Alexa Fluor 488 goat anti-rat IgG (1:200 dilution, ab150165, Abcam, America), Alexa Fluor 647 donkey anti-rabbit IgG (1:200 dilution, ab150075, Abcam, America), and Alexa Fluor 647 goat anti-mouse IgG (ab150115, Abcam, America, 1:200 dilution).

The tissue specimens underwent deparaffinization in xylene followed by sequential rehydration through graded ethanol series. Antigen retrieval was performed by heat-mediated epitope unmasking at 95 °C for 30 min in a retrieval buffer. Membrane permeabilization was achieved using 0.1% Triton X-100 in PBS for 10 min, followed by immunological blocking with 1% BSA in PBS for 60 min. The specimens were then immunolabeled with primary antibodies diluted in 1% BSA/PBS and incubated at 4 °C overnight. Subsequently, fluorophore-conjugated secondary antibodies were applied under dark conditions at ambient temperature. Nuclear counterstaining was performed using DAPI (10 μg/mL, Solarbio) before mounting with an antifading mounting medium (Solarbio). Triple PBS washing steps were implemented between each procedural stage. Immunofluorescence imaging was conducted using an Olympus FV3000 confocal microscope. The primary antibodies utilized in this study included IL-1β (1:50 dilution, 52012, Santa Cruz Biotechnology, America), TNF-α (1:50 dilution, 346654, ZEN-BIOSCIENCE, China), CD44 (1:100 dilution, 14044182, eBioscience, America), ALP (1:50 dilution, 381009, ZEN-BIOSCIENCE, China), and BMP2/4 (1:50 dilution, 137087, Santa Cruz Biotechnology, America).

## Transcriptome sequencing and data analysis

In our experimental protocol, we cultured hMSCs at a concentration of $5 \times 10^6$ cells/mL in 10 mm culture plates and administered various experimental treatments. Cell lysates were obtained using TRIzolTM reagent (15596026, Invitrogen, CA, America) and preserved at −80 °C for subsequent analysis. We assessed RNA quality parameters utilizing a NanoDrop 2000 spectrophotometer (Thermo Fisher Scientific, Wilmington, DE) for concentration and purity measurements, while RNA integrity was verified using the RNA Nano 6000 Assay Kit with the Agilent Bioanalyzer 2100 system (Agilent Technologies, CA, America). Following standard protocols, we performed library preparation and sequencing on the Illumina NovaSeq platform, generating 150 bp paired-end reads. For differential gene expression analysis, we employed DESeq2_edgeR, implementing stringent selection criteria (adjusted Fold Change ≥ 1.5, FDR < 0.05). We conducted comprehensive bioinformatics analyses through the BMKCloud platform (www.biocloud.net), encompassing GO term enrichment, PCA, KEGG classification and pathway enrichment, and GSEA (www.biocloud.net).

## Comet assays

DNA damage assessment was performed utilizing single-cell gel electrophoresis (comet assay) following the manufacturer's protocol (Comet Assay Kit C2041S, Beyotime, China). hMSCs were suspended in PBS at a concentration of $1 \times 10^6$ cells/mL, and subsequently, 10 μL of the cell suspension was combined with 75 μL of pre-warmed (37 °C) low-melting agarose in microcentrifuge tubes. The cell-agarose mixture (70 μL) was carefully transferred onto pre-coated and pre-warmed comet slides (FSL061-25pcs, Beyotime, China), followed by an additional layer of 75 μL pre-warmed low-melting agarose after the agarose solidified. After solidification at 4 °C for 30 min, slides underwent overnight lysis at 4 °C. For neutral comet analysis, PBS-washed slides were subjected to electrophoresis in a neutral buffer (2 mM $Na_2EDTA$, 90 mM boric acid, 90 mM Tris, pH 8.5) at 0.75 V/cm for 20 min. For alkaline comet analysis, slides were equilibrated in an alkaline unwinding solution (1 mM EDTA, 200 mM NaOH, pH > 13) for 20 min at ambient temperature, followed by electrophoresis in pre-chilled alkaline buffer at 0.75 V/cm for 20 min. Nuclear DNA was visualized using Propidium Iodide staining according to the manufacturer's specifications. Fluorescence microscopy (IX73, Olympus, Tokyo, Japan) was employed for image acquisition, and quantitative analysis was conducted using the OpenComet plugin within ImageJ software.

## The establishment of ROS-associated jaw defect mice

All animal experiments and associated procedures, including euthanasia, were carried out in accordance with the protocols approved by the Institutional Animal Care and Use Committee at Sichuan University (Approval Number: WCHSIRB-D-2020-361). The study received further approval from the Laboratory Animal Welfare and Ethics Committee of West China Hospital of Stomatology, ensuring compliance with the ARRIVE guidelines. For this study, male C57BL/6 mice (6 weeks of age) were maintained under standardized laboratory conditions with circadian entrainment (12:12 h light-dark photoperiod, photophase: 8:00 a.m. to 8:00 p.m.), and provided ad libitum access to standard rodent chow (catalog #1010038, Jiangsu-Xietong, Inc., Nanjing, China) and water. Environmental parameters were regulated within optimal ranges (ambient temperature: 20–26 °C; relative humidity: 40–70%). A mandibular periapical bone defect model was established in mice. Every effort was made to minimize pain and discomfort during surgery. General anesthesia was achieved through initial inhalation of 4% (w/v) isoflurane, followed by ketamine/xylazine administration (60 mg/kg and 12 mg/kg, respectively) via intraperitoneal route, with concurrent subcutaneous buprenorphine for perioperative analgesia. Following surgical site preparation and aseptic technique, an incision was made along the inferior mandibular margin to expose the buccal cortical plate. A standardized osseous defect was created using a BR-49 round bur to penetrate the buccal cortex to full bur depth. In addition, a reactive oxygen species (ROS)-enhanced jaw defect model was established in C57BL/6 male mice through the administration of lipopolysaccharide (LPS) (1 mg/mL, 0.02 mL) dissolved in saline. Following the creation of the defect, a gelatin sponge mixed with either saline or 100 μg/mL of artificial biocatalysts was applied to fill the defect and promote wound healing.

## The treatment of ROS-associated jaw defect mice

To evaluate the therapeutic efficacy of biocatalysts, mice were allocated into four experimental groups. Group 1, serving as the control, included jaw defect mice treated with saline ($n = 3$). Group 2, termed the ROSup group, consisted of mice with ROS-associated jaw defects subjected to LPS treatment ($n = 3$). Group 3, labeled the hydroxide-ROSup group, involved ROS-associated jaw defect mice treated with 100 μg/mL hydroxide dissolved in saline ($n = 3$). Group 4, named the Ru-hydroxide-ROSup group, included ROS-associated jaw defect mice treated with 100 μg/mL Ru-hydroxide in saline ($n = 3$). The mice were sacrificed at 2-, 4-, and 8-weeks post-surgery, at which points tissue samples were collected for histological, immunohistochemical, immunofluorescence, and Micro-computed tomography (μ-CT) analyses. Furthermore, the body weights of the mice were monitored daily for 14 consecutive days.

## Inductively coupled plasma mass spectrometry (ICP-MS) detection

The concentration of released metal ions and material degradation kinetics were quantitatively assessed via ICP-MS. The degradation profile was evaluated by immersing 1 mg of Ru-hydroxide in 10 mL of PBS (pH 7.4, $n = 1$) or 10 mL of cell culture medium (HUXMX-90021, Cyagen, China, $n = 1$). The temporal degradation behavior was monitored at predetermined intervals of 7, 14, 21, and 28 days. Post-incubation, the samples underwent centrifugation at $12,000 \times g$ for 20 min to separate the particulate matter. The resultant supernatant was subsequently analyzed using an 7850 ICP-MS (Agilent, China) system to determine the ionic composition and degradation parameters. For mass analysis, the instrument was calibrated with the following settings: power at 1600 W, radio frequency matching at 1.8 V, peristaltic pump speed at 0.1 rps, atomizing gas flow rate at 1.05 L/min, auxiliary gas flow rate at 0.9 L/min, plasma gas flow rate at 15 L/min, sampling depth at 10 mm, and atomizing chamber temperature at 2 °C. The

sampling period for the mass spectrum is 0.31 s, with an integration time of 0.1 s. The concentrations of elements in the test solution, reported in mg/L, are directly obtained from the instrument testing. The collected data are processed using Origin 2022.

### In vivo toxicity assessment

To assess the systemic toxicological profile of synthetic biocatalytic constructs, comprehensive histopathological and hematological analyses were conducted on male C57BL/6 murine models following a 2-week post-operative period. Major organ systems, specifically heart, liver, spleen, lung, and kidneys, were harvested and subjected to H&E staining protocols for microscopic evaluation of tissue architecture and cellular morphology. A concurrent quantitative assessment of hematological parameters was performed to evaluate systemic responses.

### ROS determination in vivo

Unfixed mandible cryosections were freshly prepared and incubated with Dihydroethidium (S0063, Beyotime, China) for 20 min under dark conditions. After incubation, a confocal laser scanning microscope (FV3000, Olympus, Tokyo, Japan) was employed to detect and visualize ROS.

### Micro-computed tomography analysis

The mandibles were preserved in 10% formalin before undergoing analysis. μ-CT imaging was performed using a μ-CT Scanner (μ-CT50, Scanco, Bassersdorf, Zurich, Switzerland) under specific settings: 165 μA current, 60 kV voltage, 450 ms exposure time, and a resolution of 10 μm. Bone parameters, including Tb.N, Tb.Th, BV/TV, and BMD, were assessed based on the standardized nomenclature provided by the μ-CT scanner protocol to evaluate the microstructural properties.

### Cell apoptosis

For hMSCs adhesion, they were plated in 6 mm dishes. Following exposure to various experimental conditions for 1 h, the cells were maintained in a fresh culture medium for an additional 24 h period. The culture supernatant was harvested, and cells were subjected to dual PBS washing steps, retaining the wash solutions for further analysis. The cells were then digested using 0.25% trypsin solution (EDTA-free), and the resulting cell suspension was transferred to a centrifugation tube. The previously collected supernatant and wash solutions were mixed with the cell suspension. The amalgamated solutions underwent centrifugation at $300 \times g$ for 5 min, followed by supernatant removal. The cellular pellet was resuspended in PBS and subjected to an additional centrifugation step under identical conditions. Apoptotic analysis was performed using the Annexin V-FITC/PI Apoptosis Detection Kit (AD10, Dojindo, Japan). Cells were resuspended in $1 \times$ Annexin V Binding Solution to achieve a final concentration of $1 \times 10^6$ cells/mL. A 100 μL volume of cell suspension was supplemented with 5 μL each of Annexin V-FITC conjugate and PI Solution, followed by a 15-min dark incubation at ambient temperature. After incubation, 400 μL of $1 \times$ Annexin V Binding Solution was added, and the samples were analyzed within 1 h. Cell populations were categorized into four quadrants based on fluorescence intensity, corresponding to live, early apoptotic, late apoptotic, and necrotic cells, while small-sized cells and particles were excluded through specific gating strategies. The resulting figures were generated using Flowjo software (version 10.8.1), with the gating strategies illustrated in Supplementary Figs. 33, 34.

### Osteogenesis differentiation

Cells were maintained in osteogenic media enriched with 10 mM β-glycerophosphate (Sigma-Aldrich, the United States), 50 μg/mL L-ascorbic acid (Sigma-Aldrich, the United States), and 10 nM dexamethasone (Sigma-Aldrich, the United States). The samples were washed with PBS, followed by fixation in 4% paraformaldehyde for 15 min on the third day of culture. The activity of alkaline phosphatase (ALP) was then measured using a kit from Beyotime (China). Alizarin Red staining was performed to assess calcium deposits in the fixed samples on day 21, and images were obtained using a stereomicroscope (SZX16, Olympus, Japan).

### Statistical analysis

Statistical analysis was performed using GraphPad Prism 8.0 software (GraphPad Software Inc.), and figures were generated with Origin 2022b. Image analysis for both in vitro and in vivo experiments was conducted using ImageJ and Image-Pro Plus software. Experimental parameters, including sample size ($n$), statistical significance ($p$), data normalization protocols, and specific statistical tests, are comprehensively detailed in the corresponding figure legends. Quantitative data were collected at least three independent times. All data are expressed as the mean values ± SD. Statistical significance was calculated using the two-tailed Student's $t$-test or one-way analysis of variance (ANOVA); all tests were two-sided. Statistical significance was set at $*p < 0.05$, $**p < 0.01$, $***p < 0.001$, $^{\#}p < 0.05$, $^{\#\#}p < 0.01$, $^{\#\#\#}p < 0.001$, and ns represents no significant difference.

### Reporting summary

Further information on research design is available in the Nature Portfolio Reporting Summary linked to this article.

## Data availability

The main data supporting the results of this study are available within the paper and its Supplementary Information. Raw RNA sequencing data generated in this study have been deposited in the NCBI SRA database under accession number GSE235884. Source data are provided with this paper.

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

## Acknowledgements

This work was financially supported by the National Key Research and Development Program of China (2024YFE0201200 [C.C.], 2023YFC3605600 [L.Y.]), the National Natural Science Foundations of China (52173133 [C.C.], 52161145402 [C.C.], 52373148 [C.C.], 82470962 [M.R.B.], U21A20368 [L.Y.]), Sichuan Science and Technology Program (2024NSFSC0672 [M.R.B.], 2023YFS0019 [L.Y.]), the State Key Laboratory of Polymer Materials Engineering (sklpme2021-4-02 [C.C.]), the 1·3·5 Project for Disciplines of Excellence, West China Hospital, Sichuan University (ZYJC21047 [C.C.]), Research Funding from West China School/Hospital of Stomatology Sichuan University (RCDWJS2023-16 [M.R.B.]), Research and Develop Program, West China Hospital of Stomatology Sichuan University (RD-02-202206 [M.R.B.]), and Med-X Innovation Program of Med-X Center for Materials, Sichuan University

(MCMGD202301 [L.Y.]). Prof. Mohsen Adeli would like to thank the Iran Science Elites Federation and Iran National Science Foundation (Project Number 4001281) for financial support. We gratefully acknowledge Dr. Mi Zhou and Dr. Chao He for their analytical support. We also thank Qiang Guo (State Key Laboratory of Oral Diseases, West China Hospital of Stomatology, Sichuan University) for characterizing micro-CT and Ning Gi, Yuwen Luo, Xuelin Huang (State Key Laboratory of Oral Diseases, West China Hospital of Stomatology, Sichuan University) for animal care and Wenlin Chu (State Key Laboratory of Oral Diseases, West China Hospital of Stomatology, Sichuan University)for flow cytometry analysis.

## Author contributions

T.W. and M.R.B. contributed equally to this work. T.W. and M.R.B. performed the experiments and analyzed the results. T.W., M.R.B., W.G., and M.A. assisted with the figure production and experimental design. T.W., M.R.B., W.G., L.Y., and C.C. wrote the manuscript. C.C. designed the experiments, corrected the manuscript, and supervised the whole project. All authors discussed the results and commented on the manuscript.

## Competing interests

The authors declare no competing interests.
