## [Transparent Peer Review file · Nature Communications]

Bioinspired artificial antioxidantases for efficient redox homeostasis and maxillofacial bone regeneration

Corresponding Author: Professor Chong Cheng

Version 0:

Reviewer comments:

Reviewer #1

(Remarks to the Author)

This study addressed an interesting topic in inflammatory maxillofacial defects regeneration. Inspired by intracellular antioxidant defense systems, the authors designed an efficient artificial antioxidantase via using Ru-doped layered double hydroxide for superior redox homeostasis and maxillofacial bone regeneration. My recommendations for minor revisions are as follows:

1. Why did the authors choose element Ru? What is the superiority of this element? As we know that Ru is relatively rare in nature, is there any possibility to extend this design to other elements? It might be better to introduce this background for readers not familiar with this field.
2. Did the authors compare MSC protection effect of Ru-hydroxide with other ROS elimination materials? Though there is a comparison of parameters V_{max} and TON among different materials in Figure 2e, we don't know to what extent do these parameters accurately reflect its efficacy in protecting MSCs and enhancing bone regeneration in an inflammatory environment.
3. The authors have published another ROS elimination material RuNC-CN, what is the main difference between these two strategies? What advantages do the materials in this study offer compared to RuNC-CN?

Reviewer #2

(Remarks to the Author)

In the realm of regenerative medicine, designing biocompatible materials that can orchestrate redox homeostasis is of paramount importance for the advancement of tissue repair and regeneration. The present study synthesizes a new artificial antioxidantase, Ru-hydroxide, which exhibits exceptional capabilities in maintaining redox balance and promoting maxillofacial bone regeneration. Interestingly, the authors have elucidated that the bioinspired proton/electron transfer catalytic mechanism is pivotal for reactive oxygen species (ROS) biocatalysis. The study further demonstrates that Ru-hydroxide not only sustains the viability and proliferation of endogenous stem cells but also facilitates their osteogenic differentiation, thereby fulfilling the clinical requirements for maxillofacial bone repair. This research opens a promising strategy in regenerative medicine, with implications for the treatment of a variety of inflammation-related diseases, thus holding immense potential in the context of biomaterials, regenerative medicine, and future clinical applications. The manuscript is commendably crafted, showcasing a high degree of novelty and clinical relevance. Therefore, I recommend its publication in Nature Communications after addressing the following concerns.

1. In Figure 1e, the authors observed that the Ru 3p peak of Ru-hydroxide shifted by about 0.8 eV, indicating a significant electron transfer from the hydroxide substrate to the Ru atoms. However, the authors did not analyze the shifts in the Co 2p and Ni 2p peaks in Ru-hydroxide to determine if there was any electron transfer to the Ni and Co surfaces, thereby creating new active centers.
2. Despite the catalyst's remarkable resilience to H₂O₂-induced poisoning, its durability in the presence of superoxide anions ($\cdot\text{O}_2^-$) warrants further investigation.
3. The observed higher Michaelis constant (K_m) for Ru-hydroxide relative to Ru-oxide, which may suggest an augmented desorption capacity on substrates, calls for a deeper theoretical elucidation to substantiate this hypothesis.
4. In Figures 3a and 3b, the authors demonstrated through charge density plots and Bader charge analysis that the Ru

atoms in Ru-hydroxide lost fewer electrons compared to the Ru atoms in Ru-oxide. However, the effectiveness of Bader charge analysis is poor for such a complex system. It is recommended to calculate Hirshfeld charges for a more accurate charge analysis.

5. In Figure 3d, the authors proposed different mechanisms for the catalase-like reaction between Ru-hydroxide and Ru-oxide. However, the authors did not analyze the transition states. For example, in Ru-hydroxide, transition states occur between S1-S2 and S4-S5, which are critical and can significantly impact the rate-determining step (RDS). Additionally, for the S6 state of Ru-hydroxide, how to directly desorb OOH into slab is not explained by the authors.

6. The authors did not label the orbitals corresponding to the projected density of states (PDOS) in Figure 3f, nor did they analyze the PDOS before and after OOH adsorption. Without this analysis, the figure lacks significance in the context of the study.

7. The pioneering strategy of mimicking intracellular antioxidant defense systems to fabricate antioxidative materials endowed with proton/electron transfer capabilities marks a significant conceptual leap. While the proton transfer pathway delineated in Figure 3 has been rigorously confirmed, more clarification is required to elucidate the electron transfer processes.

8. Does the hydroxylation methodology present a scalable solution for the broader class of Ru-based materials?

9. In-situ experimentation stands as a valuable technique for substantiating catalytic mechanisms; however, the authors are encouraged to delineate the specific procedural details of the in-situ assays.

10. The inclusion of unstained control data in the supplementary materials is imperative to corroborate the threshold selection in apoptosis flow cytometry analyses.

11. Figure 5 illustrates the beneficial effects of Ru-hydroxide on human mesenchymal stem cells (hMSCs) across a spectrum of cellular functions, inferred through KEGG pathway analysis in the context of H₂O₂ and Ru-hydroxide treatments. Nevertheless, the authors should anchor these inferences with pertinent literature to establish the nexus between the observed cellular functions and the implicated pathways.

12. The colour fill in figure 6g is different from the other figures, please standardise.

Reviewer #3

(Remarks to the Author)

In this article, Wang and colleagues evaluated the use of novel ROS inhibitor/quencher called Ru-doped layered double hydroxide (Ru-hydroxide) for superior redox homeostasis and maxillofacial bone regeneration. Specifically, Ru-hydroxide attenuates ROS in stem cells and decreases DNA damage and increases the stem cell potential. Further, Ru-hydroxide can effectively sustain stem cell viability, proliferation and osteogenic differentiation in elevated ROS environments. It also showed in-vivo biocatalytic microenvironment modulations during inflammatory bone tissue regeneration. Overall the manuscript is well executed, however, there are some queries and concerns authors need to address for the improvement/clarifications that supports the conclusions.

- While Ru-Hydroxide shows to deplete ROS concentration in stem cells, how will it achieve the optimal concentration of ROS? Mainly because low level ROS is also important for stem cells to maintain quiescence and self-renewal (PMID: 24974178).
- Bone healing is a month to year long process. How stable is the Ru-hydroxide? will the new stem cells that arise at the end of bone reformation will have similar access to Ru-hydroxide in-vivo compared to the stem cells that forms early in the repair?
- What happens to non-stem cells that picks up Ru-hydroxide?
- Gamma-H2AX measures indirect double strand breaks and is highly variable based on the cell type or the time we fix these cells. For a more prominent study like this, it is important that the authors evaluate these DNA damage responses using more relevant assays like alkaline and neutral COMET assays for single and double strand breaks, oxidative DNA damage/8-Oxoguanine, other signal proteins modified due to free radicals and/or Chromosomal aberrations.
- Figure 6B: Treatment with Ru-hydroxide reduces the levels of TNF-alpha and IL-1B well beyond the control levels. Again, low levels of TNF-alpha and IL-1B is also required to maintain the normal cellular homeostasis. Especially, TNF-alpha is required for normal hematopoiesis and will Ru-hydroxide impact the normal hematopoiesis in maxillofacial regions?
- Most of the DNA damage response pathways were analyzed using genomic studies in the current study. However, most of the DNA damage responses are modified post-transcriptionally at the protein levels. Analysis of some of these proteins and their posttranslational modifications are as important as the genomic data about DNA damage responses as the authors have described in the article.

Version 1:

Reviewer comments:

Reviewer #1

(Remarks to the Author)

The revised version is found to be satisfactory.

Reviewer #2

(Remarks to the Author)

The authors have addressed all my concerns. It can be accepted by Nature Communications.

Reviewer #3

(Remarks to the Author)

Authors adequately addressed the reviewer queries with additional experimental data. The study is detail and meticulously executed. The data supports the conclusions.

Point-by-point response to the detailed comments by reviewers of “*Bioinspired artificial antioxidases with rapid and sustained proton/electron transfer for superior redox homeostasis and maxillofacial bone regeneration*” with manuscript ID: NCOMMS-24-36973.

REVIEWER COMMENTS

Reviewer #1 (Remarks to the Author):

“This study addressed an interesting topic in inflammatory maxillofacial defects regeneration. Inspired by intracellular antioxidant defense systems, the authors designed an efficient artificial antioxidase via using Ru-doped layered double hydroxide for superior redox homeostasis and maxillofacial bone regeneration. My recommendations for minor revisions are as follows:”

Response to the general comment:

Thank you for your insightful and positive comments of our study on inflammatory maxillofacial defect regeneration. Based on your comments and the suggestions from other reviewers, we have conducted more systematic experiments and refined the content throughout the manuscript. All necessary data have been added to support our claims, and we have thoroughly addressed all questions and concerns in the revised manuscript and supplementary information. Therefore, we believe that the quality of this paper has been significantly enhanced, and we thank you once again for your helpful suggestions and considerable efforts.

(1) Why did the authors choose element Ru? What is the superiority of this element? As we know that Ru is relatively rare in nature, is there any possibility to extend this design to other elements? It might be better to introduce this background for readers not familiar with this field.

Response to comment:

Thanks for your important and helpful comments on improving the quality of our manuscript. We agree that the introduction of foundational background information on ruthenium (Ru) is essential for a comprehensive understanding of our study.

The initial discovery of Fe₃O₄ as an artificial enzyme material, reported by researchers in 2007 (*Nat. Nanotechnol.*, **2007**, 2, 577-583), sparked significant interest in the application of metal oxides based on iron (Fe) in enzyme-like catalysis. Since then, various artificial enzymes and enzyme-like structures utilizing elements, such as Fe, cobalt (Co), and manganese (Mn), have been developed (*Adv. Mater.*, **2016**, 28, 1387-1393; *Analyst*, **2012**, 137, 4552-4558; *Chem. Commun.*, **2012**, 48, 2540-2542). While these materials exhibit some degree of enzyme-like catalytic activities, they typically suffer from low efficiency, which hinders their practical applications. Ru, as a member of the iron group elements, offers unique physicochemical properties and advantages. First and foremost, Ru metals exhibit good biocompatibility, making it suitable for biologically relevant catalytic reactions and biomedical applications (*ACS Nano*, **2023**, 17, 16501; *Angew. Chem. Int. Ed.*, **2024**, 63, e202405679; *Adv. Funct. Mater.*, **2024**, 34, 2315885). Second, Ru possesses a higher number of *d*-electrons compared to Fe, which is vital for effective bond formation during biocatalysis (*Adv. Mater.*, **2022**, 34, 2206208). This unique electronic structure allows Ru to participate in catalysis with greater efficiency, thereby boosting the catalytic performances. Furthermore, Ru features a sufficient number of unoccupied orbitals, facilitating the coordination of reaction intermediates and further accelerating the catalytic process (*Nat. Commun.*, **2024**, 15, 5419; *Nat. Commun.*, **2019**, 10, 4936). Another notable benefit of the Ru metal site is its excellent redox stability (*Chem*, **2023**, 9, 1882-1896; *Angew. Chem. Int. Ed.*, **2024**, 63, e202317220), which positively influences the kinetics of adsorption and dissociation during catalysis, leading to more efficient and stable catalytic processes. As a result of these advantageous physicochemical characteristics, it is suggested that Ru metal sites may display more favorable catalytic properties compared to the earlier reported Fe metal sites in enzyme-like biocatalytic reactions, thereby we try to use the Ru metal species to construct a new type of efficient and biocompatible artificial antioxidantases for inflammatory maxillofacial defects. We appreciate your encouragement to highlight Ru's advantages, and we have elaborated on these points more clearly in our revised manuscript, as also outlined below:

Page 4 in the revised manuscript: “Evolution has led to the development of enzymes that typically incorporate the transition metal species, such as iron (Fe), copper, and manganese^{57,58}. Ruthenium (Ru), as a member of iron group elements, presents advantageous biocompatibility and unique metallic properties^{59,60}. In comparison to Fe species, the Ru features more *d*-electrons, sufficient unoccupied orbitals, and superior redox stability, which benefit bond formation, reactive intermediate coordination, and adsorption-dissociation dynamics during catalysis^{61,62}. Therefore, constructing Ru-based artificial antioxidantases with rapid proton and electron transfer mechanisms presents a promising avenue for enhancing the catalytic ROS-elimination activities and supporting redox homeostasis.”

In addition, for your comment on extending this design to other elements, yes, this design can be extended to some noble metal species. During the preliminary experiment stage, we have investigated several alternative noble metal elements, particularly osmium (Os, another member of iron group elements) and rhodium (Rh, a noble metal element that is next to Ru). Both Os and Rh elements exhibited a certain degree of biocatalytic ROS scavenging properties that are worthy of further exploration. However, when selecting the final noble metal species for our experiments, we had to comprehensively consider several critical factors, including biocompatibility, multiple activities in artificial antioxidantases, and the specific therapeutic objectives of our study. After a thorough evaluation and comparison of different noble metal elements, we opted to focus on Ru as the optimal element for our research. Meanwhile, the Os and Rh may show higher potential in other biocatalytic applications. For instance, Os and Rh-based materials show higher oxidase-like properties than Ru, which may be promising for bacterial disinfection in wound healing. Consequently, we are confident that there will be opportunities to extend this design to other noble metal elements, and we are planning to investigate the potential application of Os and Rh-based materials in other fields. We are grateful for your helpful suggestions, which will undoubtedly contribute to enhancing our study.

(2) *Did the authors compare MSC protection effect of Ru-hydroxide with other ROS elimination materials? Though there is a comparison of parameters V_{max} and TON among different materials in*

Figure 2e, we don't know to what extent do these parameters accurately reflect its efficacy in protecting MSCs and enhancing bone regeneration in an inflammatory environment.

Response to comment:

Thank you for your valuable comments. In this paper, we have conducted a thorough comparison of the protective effects of Ru-hydroxide on mesenchymal stem cells (MSCs) with some other ROS elimination biocatalysts, specifically focusing on several key materials, including metal oxide, metal hydroxide, and Ru-oxide. These materials share high physicochemical similarities regarding material density, degradability, porosity, and biocompatibility, thus providing a robust foundation for evaluating their capacity to protect MSCs via ROS scavenging. Our findings indicate a positive correlation between the antioxidative activity of the materials and their protective effects on MSCs. However, for different types of materials, the comparative evaluation on the protective efficacy of MSCs may be significantly affected by their diverse physicochemical characteristics, such as the disparities in density, degradability, porosity, cytotoxicity, and size, which makes it difficult to achieve a unified or standardized comparison. Therefore, our paper does not include the comparisons of a broader range of different types of materials. Furthermore, during the preliminary experiment stage, we conducted a detailed comparison of the Ru-hydroxide material on biocatalytic performances with those previously reported. Under consistent dosage and operational conditions, Ru-hydroxide demonstrated superior ROS scavenging performances. Our *in vitro* assessment of antioxidative activity, as illustrated in Table 1 of the revised supplementary information, reveals that the Ru-hydroxide material exhibits optimal antioxidative activity, thereby theoretically supporting its superiority in protecting cells. Looking ahead, we plan to carry out systematic and comprehensive analyses of different types of materials or materials with varying antioxidative mechanisms to explore the differences in their efficacy and their specific mechanisms for cell protection.

In response to your comment regarding the relationship between kinetic parameters and the protective efficacy of materials for MSCs in inflammatory environments, based on our earlier findings and preliminary experimental data, it is believed that an increase in kinetic parameters, such as V_{\max} and TON, correlates positively with the materials' capacity to safeguard MSCs and enhance bone regeneration. Theoretically, an increase in these kinetic parameters suggests a corresponding

enhancement in the antioxidative properties and protective activities of these materials on MSCs. However, we agree with the reviewer that it is important to note that the microenvironments of inflammatory tissues are complex and variable during actual anti-inflammatory treatments, which encompass not only ROS but also inflammatory factors, immune cells, and various biochemicals. These elements may also result in a situation where kinetic parameters do not consistently exhibit a linear relationship with the actual anti-inflammatory effects and the promotion of bone regeneration. For instance, porous nanomaterials may demonstrate superior adsorption capacities for inflammatory factors, potentially influencing their therapeutic efficacy. Consequently, we suggest that the kinetic parameters associated with the antioxidative activities of the materials may partially reflect their ability to protect MSCs and bone regeneration-promoting capabilities. Related studies in tissue regeneration are still nascent, highlighting the need for considerable research to further explore the mechanisms by which these materials operate in intricate biological environments and their prospective clinical applications. We anticipate that future investigations will elucidate the interactions between performance parameters and biological efficacy, thereby providing both theoretical support and practical guidance for the development of more effective therapeutic strategies.

(3) *The authors have published another ROS elimination material RuNC-CN, what is the main difference between these two strategies? What advantages do the materials in this study offer compared to RuNC-CN?*

Response to comment:

We sincerely thank you for your insightful comments, which further highlight the strengths of our material design. While some earlier studies have reported the design of artificial enzyme materials with antioxidative activities, most of them show relatively low ROS eliminating activity (with a turnover number (TON) value below 1 s^{-1}), often necessitating large doses for effective therapeutic applications (*ACS Nano*, **2020**, 14, 4383; *Angew. Chem. Int. Ed.*, **2020**, 59, 9491; *Chem. Commun.*, **2018**, 55, 159; *Nat. Commun.*, **2020**, 11, 357). This underscores the urgent demand for designing new, efficient, and versatile antioxidase-like biocatalytic materials.

In our earlier research, we have discovered that a rational design of the coordination structure could enhance the intrinsic catalytic activity of metal sites (*Nature Communications*, 2021, 12, 6143. *Advanced Materials*, 2022, 34, 2108646), thus opening a new avenue for developing efficient antioxidase-like biocatalysts for various chemical/biochemical reactions. Based on the coordination strategy, our team has successfully synthesized the Ru-N coordinated artificial biocatalysts on carbon nitride (CN) frameworks with synergetic Ru clusters (Ru_{NC}-CN) for effective ROS elimination (*Adv. Mater.*, **2022**, 34, 2206208). However, despite the beneficial energy band structure of the CN substrate, which enhances the electrical conductivity and reaction kinetics on ROS elimination, its poor degradability presents a biosafety challenge for long-term use in tissue regeneration.

Designing antioxidase-like materials that combine high ROS-eliminating activity and good degradability is undoubtedly a challenging task, yet it is highly desired for broad biomedical applications. Earlier studies have indicated that metal oxides may serve as a potential substrate to incorporate Ru to create efficient artificial antioxidases. However, it is important to note that in metal oxide-based artificial antioxidases, the catalytic metal atoms are typically coordinated with electronegative oxygen elements (*Adv. Mater.*, **2022**, 34, 202207275; *J. Am. Chem. Soc.*, **2023**, 145, 19086). This bonding can impede electron transfer processes, thereby hindering the efficiency of the multi-electron reactions involved in ROS scavenging. Additionally, addressing the proton transfer associated with the efficient and stable scavenging of various types of ROS proves to be challenging.

Therefore, in this work, to protect endogenous stem cells and promote bone regeneration in inflammatory maxillofacial defects, we have proposed the *de novo* bioinspired design of an efficient artificial antioxidase using Ru-doped layered double hydroxide (named Ru-hydroxide) to achieve highly efficient multifunctional ROS scavenging activities while ensuring excellent degradability for superior redox homeostasis and maxillofacial bone regeneration. Specifically, compared to the earlier reported Ru_{NC}-CN and other Ru-based materials, the unique advantages of Ru-hydroxide include: 1) the innovative hydroxyl-synergistic monoatomic Ru centers within the Ru-hydroxide, which facilitate rapid and sustained proton/electron transfer, thereby enhancing ROS elimination activities; 2) the high electron density of the Ru centers, which optimizes the binding strength with oxygen species and enhances the resistance to H₂O₂ and •O₂⁻ toxicity; and 3) the hydroxides' inherent good degradability, low long-term cytotoxicity, and high biosafety. By leveraging these distinctive

advantages, the material developed in our study achieves a balance of high activity and degradability, which translates to enhanced repair functions and superior biosafety in inflammatory maxillofacial bone defects.

Reviewer #2 (Remarks to the Author):

“In the realm of regenerative medicine, designing biocompatible materials that can orchestrate redox homeostasis is of paramount importance for the advancement of tissue repair and regeneration. The present study synthesizes a new artificial antioxidantase, Ru-hydroxide, which exhibits exceptional capabilities in maintaining redox balance and promoting maxillofacial bone regeneration. Interestingly, the authors have elucidated that the bioinspired proton/electron transfer catalytic mechanism is pivotal for reactive oxygen species (ROS) biocatalysis. The study further demonstrates that Ru-hydroxide not only sustains the viability and proliferation of endogenous stem cells but also facilitates their osteogenic differentiation, thereby fulfilling the clinical requirements for maxillofacial bone repair. This research opens a promising strategy in regenerative medicine, with implications for the treatment of a variety of inflammation-related diseases, thus holding immense potential in the context of biomaterials, regenerative medicine, and future clinical applications. The manuscript is commendably crafted, showcasing a high degree of novelty and clinical relevance. Therefore, I recommend its publication in Nature Communications after addressing the following concerns.”

Response to the general comment:

We sincerely appreciate your recognition of our antioxidantase-like materials as highly promising biomaterials for translational and clinical applications. Your comments and suggestions are essential for us in enhancing the quality of this work. In response to your insights, we have thoroughly revised the manuscript; all queries and concerns have been addressed in the revised manuscript and supplementary information. Your guidance has been invaluable in refining our study, and we are

confident that these revisions have substantially improved the paper. Once again, we extend our thanks for your insightful suggestions and comments.

(1) In Figure 1e, the authors observed that the Ru 3p peak of Ru-hydroxide shifted by about 0.8 eV, indicating a significant electron transfer from the hydroxide substrate to the Ru atoms. However, the authors did not analyze the shifts in the Co 2p and Ni 2p peaks in Ru-hydroxide to determine if there was any electron transfer to the Ni and Co surfaces, thereby creating new active centers.

Response to comment:

We sincerely appreciate your valuable comments and constructive feedback, which have played an essential role in improving our manuscript. Here, we would like to respond to your comments in two parts comprehensively:

1) Regarding your inquiry regarding the shifts in the Co 2p and Ni 2p peaks in Ru-hydroxide: we have conducted an X-ray photoelectron spectroscopy (XPS) analysis to investigate these shifts. Our findings revealed that the Co 2p peak shifts 0.16 eV toward higher binding energy, while the Ni 2p peak shifts by 0.08 eV, indicating that the hydroxide substrate experiences electron loss as electrons are transferred to Ru (Supplementary Fig. 10). The elevated electron density of the Ru centers optimizes the binding strength of oxygen species and enhances the resistance to H₂O₂ toxicity for Ru-hydroxide, thereby facilitating effective ROS biocatalysis. The corresponding details have been shown in the revised manuscript and revised supplementary information, as also outlined below:

Page 6 in the revised manuscript: “Notably, the Ru 3p peaks of Ru-hydroxide downshift by about 0.8 eV compared to Ru-oxide, while the Co 2p peaks and Ni 2p peaks of Ru-hydroxide shift to a higher field compared to hydroxide (Supplementary Fig. 10). These results imply a pronounced electron transfer from the hydroxide substrate to the Ru atoms, thus resulting in Ru sites with high electron density for efficient ROS biocatalysis.”

2) Regarding your question about the generation of new active centers: it is our belief that, although the electronic structure of Co and Ni is modified upon the doping of Ru, such changes

support the catalytic activity of Ru by donating electrons, with no new active sites being created. We employed density functional theory (DFT) calculations to assess the reactivity of various metal centers, including Ru, Co adjacent to Ru, and Ni adjacent to Ru. The results indicate that the Ru site exhibits higher electrophilicity compared to Co and Ni, suggesting that Ru is the primary active center responsible for interacting with oxygen species during catalysis (Supplementary Fig. 17). Overall, the exceptional catalytic performance of our Ru-hydroxide is attributed to the synergistic interaction between the hydroxide substrate, which supplies both electrons and protons and the Ru active site, which reacts with oxygen species in the catalytic process. The corresponding details have been added to the revised manuscript and revised supplementary information, as also shown below:

Page 11 in the revised manuscript: “Primarily, the reactivity of various metal sites in Ru-hydroxide was assessed through Hirshfeld charge analysis. The results indicate that Ru centers exhibit higher electrophilicity, with an electron density of $-0.542 |e|$, in comparison to Co sites ($-0.198 |e|$) and Ni sites ($-0.180 |e|$), suggesting that Ru centers are more inclined to interact with oxygen species during catalysis (Supplementary Fig. 17).”

Supplementary Fig. 10. **a** Co 2p and **b** Ni 2p XPS spectra of Ru-hydroxide and hydroxide. Sat. indicates the satellite peaks. Source data are provided as a Source Data file.

Supplementary Fig. 17. Hirshfeld charge analysis of Ru, Co, Ni species in Ru-hydroxide. Source data are provided as a Source Data file.

(2) *Despite the catalyst's remarkable resilience to H₂O₂-induced poisoning, its durability in the presence of superoxide anions ($\cdot\text{O}_2^-$) warrants further investigation.*

Response to comment:

Thank you for your invaluable comments and suggestions on how to enhance the quality of our manuscript. We appreciate your emphasis on the need to investigate the durability of catalysts for eliminating superoxide anions ($\cdot\text{O}_2^-$). In response to your recommendations, we have added cyclic stability tests specifically designed to assess the catalyst's capacity to remove $\cdot\text{O}_2^-$. The results were promising; the Ru-hydroxide material exhibited robust activity in $\cdot\text{O}_2^-$ removal even after six cycles (Supplementary Fig. 12). This finding not only substantiates the catalyst's durability but also highlights its effectiveness against broad types of ROS, thereby addressing the concerns raised in your comment. Again, Thank you for your constructive input, which has been invaluable in refining our work. We have included a discussion of the cyclic stability tests in the revised manuscript and revised supplementary information, as also shown below:

Page 9 in the revised manuscript: “Primarily, the biocatalytic conversion of $\bullet\text{O}_2^-$ (SOD-mimics) into H_2O_2 and O_2 by Ru-hydroxide was studied, which demonstrates its superior efficacy in $\bullet\text{O}_2^-$ elimination compared to Ru-oxide, hydroxide, and oxide (Fig. 3a). In-depth studies have been performed to assess the long-term activity and stability of Ru-hydroxide as a SOD-like antioxidant; the data indicates that Ru-hydroxide can sustain its high activity with no apparent reduction in performance over the course of six cycles (Supplementary Fig. 12).”

Supplementary Fig. 12. Stability test of Ru-hydroxide to eliminate $\bullet\text{O}_2^-$. Source data are provided as a Source Data file.

(3) The observed higher Michaelis constant (K_m) for Ru-hydroxide relative to Ru-oxide, which may suggest an augmented desorption capacity on substrates, calls for a deeper theoretical elucidation to substantiate this hypothesis.

Response to comment:

We sincerely appreciate your insightful and constructive comments aimed at enhancing the quality of our manuscript. As you pointed out, the higher Michaelis constant (K_m) for Ru-hydroxide compared to Ru-oxide suggests an increased desorption capacity on the substrates. In response to

your observations, we have supplemented our discussion with theoretical calculations to further substantiate this finding. We calculated the adsorption free energy for H₂O₂ on Ru sites (Supplementary Fig. 14). Our findings reveal that Ru-hydroxide displays weaker interactions with H₂O₂ due to its elevated electron density compared to Ru-oxide. Furthermore, the Gibbs free energy diagram illustrates that Ru-hydroxide interacts less strongly with various oxygen species, including H₂O₂, than Ru-oxide (Fig. 4d). This observation implies that Ru sites in Ru-hydroxide successfully optimize their binding interactions with oxygen species, leading to improved substrate desorption and, consequently, increased antioxidase-like kinetics of Ru-hydroxide. The corresponding details have been added to the revised manuscript and revised supplementary information, as also shown below:

Page 9 in the revised manuscript: “The Ru-hydroxide demonstrates a notably higher K_m compared to Ru-oxide, suggesting superior desorption capabilities on intermediates, which aligns with the results of our density functional theory (DFT) calculations (Fig. 3d, Supplementary Fig. 14, and Supplementary Table 3).”

Supplementary Fig. 14. Calculated the free energy for the adsorption of H₂O₂ by Ru-hydroxide and Ru-oxide. Source data are provided as a Source Data file.

Fig. 4 Theoretical analysis to disclose the mechanisms for superior antioxidant-like activities. **d** Gibbs free-energy diagram of H_2O_2 decomposition on Ru-hydroxide and Ru-oxide.

(4) In Figures 3a and 3b, the authors demonstrated through charge density plots and Bader charge analysis that the Ru atoms in Ru-hydroxide lost fewer electrons compared to the Ru atoms in Ru-oxide. However, the effectiveness of Bader charge analysis is poor for such a complex system. It is recommended to calculate Hirshfeld charges for a more accurate charge analysis.

Response to comment:

We sincerely appreciate your valuable feedback, which has allowed us to improve the quality of this paper. We agree with the reviewer that the limitations associated with Bader charge analysis are in the context of this complex system. In light of your suggestion, we have utilized Hirshfeld charge calculations to achieve a more accurate assessment of electron density variations between Ru in Ru-hydroxide and Ru-oxide. Our findings continue to support the conclusion that Ru atoms in Ru-hydroxide lose fewer electrons than those in Ru-oxide (Figs. 4a, b). Additionally, we conducted X-ray photoelectron spectroscopy (XPS), revealing that the Ru 3p peaks for Ru-hydroxide shift down by approximately 0.8 eV compared to Ru-oxide, further corroborating an increased electron density in Ru-hydroxide. Your insights have significantly contributed to refining our research, and we are

grateful for your suggestions. Detailed information about these modifications can be found in the revised manuscript, as also shown below:

Page 11 in the revised manuscript: “Furthermore, the Hirshfeld charges of Ru sites for the Ru-hydroxide and Ru-oxide are studied, which demonstrates that the Ru sites in Ru-hydroxide lose fewer electrons (-0.542 |e|) compared to Ru-oxide, thus exhibiting a much higher electron density than that in Ru-oxide (Figs. 4a, b).”

Fig. 4 Theoretical analysis to disclose the mechanisms for superior antioxidant-like activities.

Differential charge density analysis and Hirshfeld charge of Ru centers in **a** Ru-hydroxide and **b** Ru-oxide (yellow and cyan represent charge accumulation and depletion, respectively, with a cutoff value of $0.006 \text{ e} \cdot \text{Bohr}^{-3}$ for the density-difference isosurface). Atom colors: cyan, Ru; white, O; pink, H; wathet blue, Co; and navy blue, Ni.

(5) In Figure 3d, the authors proposed different mechanisms for the catalase-like reaction between Ru-hydroxide and Ru-oxide. However, the authors did not analyze the transition states. For example, in Ru-hydroxide, transition states occur between S1-S2 and S4-S5, which are critical and can significantly impact the rate-determining step (RDS). Additionally, for the S6 state of Ru-hydroxide, how to directly desorb OOH into slab is not explained by the authors.

Response to comment:

Thank you for your valuable insights and constructive comments, which have greatly enhanced our manuscript. We would like to apologize for the lack of a thorough analysis regarding the transition states in our earlier manuscript. To optimize our computational efficiency, we initially employed a simplified model and conducted thermodynamic calculations in our early research.

However, the investigation of transition states in the context of dynamics is highly complex and influenced by numerous factors. Here, to simulate the dynamic processes involved in the catalytic reaction, we have implemented a more complex and comprehensive model for the transition state calculations, which considers the effects of the decomposition product, H_2O , throughout the reaction. Based on this, we have included a detailed examination of the transition states, particularly those occurring between **S1-S2** and **S4-S5** in Ru-hydroxide (Figs. R1, R2). Specifically, in the transition from **S1** to **S2**, H_2O_2 undergoes homolysis to yield two $\cdot\text{OH}$ species. One $\cdot\text{OH}$ interacts with the Ru site, while the other combines with the H in the Ru-hydroxide to form $\cdot\text{H}_2\text{O}$, thus yielding the **S2** state. In the case of the transition from **S4**, H_2O_2 is adsorbed at the Ru site, leading to heterolysis that produces an $\cdot\text{H}$ species and an $\cdot\text{OOH}$ species. The $\cdot\text{H}$ species then reacts with the OH group at the Ru site to yield $\cdot\text{H}_2\text{O}$, while the $\cdot\text{OOH}$ species is further stabilized at the Ru site to form **S5**. We appreciate your thoughtful consideration, and we believe that this enhanced explanation effectively addresses your concerns and clarifies the catalytic mechanism involved.

Fig. R1. Proposed transition states occur between S1 and S2, as well as between S4 and S5, during the decomposition of H_2O_2 on Ru-hydroxide.

Fig. R2. The energy of states occurs between S1 and S2, as well as between S4 and S5, during the decomposition of H_2O_2 on Ru-hydroxide.

Regarding the desorption of $\ast\text{OOH}$ into the slab, the substantial attraction of the H vacancy on Ru-hydroxide facilitates the return of H from the $\ast\text{OOH}$ intermediate to Ru-hydroxide, resulting in the production of $\ast\text{OO}$ and ultimately leading to the release of O_2 (Fig. R3). We appreciate your thoughtful consideration, and we have provided a thorough explanation in the revised manuscript, as also shown below:

Page 11 in the revised manuscript: “It is noteworthy that the conversion of the $\ast\text{OOH}$ intermediate to generate $\ast\text{OO}$ and $\ast\text{H}$ remains a high-energy barrier in Ru-oxide, whereas it proceeds spontaneously in Ru-hydroxide due to the reprotonation process of the hydroxide substrate (Fig. 4d). In this process, the substantial attraction of the H vacancy on Ru-hydroxide facilitates the return of H from the $\ast\text{OOH}$ intermediate to Ru-hydroxide, resulting in the production of $\ast\text{OO}$ and ultimately leading to the release of O_2 (Supplementary Fig. 20).”

Fig. R3. The *OOH desorption process occurs during the H₂O₂ decomposition of Ru-hydroxide. Atom colors: cyan, Ru; white, O; pink, H.

(6) The authors did not label the orbitals corresponding to the projected density of states (PDOS) in Figure 3f, nor did they analyze the PDOS before and after OOH adsorption. Without this analysis, the figure lacks significance in the context of the study.

Response to comment:

Thank you for your helpful comments and suggestions. We wholeheartedly agree that labeling the orbitals corresponding to the projected density of states (PDOS) is essential for enhancing the clarity and significance of our manuscript. In response, we have now included orbitals for the PDOS, specifically labeling Ru as the *d* orbital and OOH as the superposition of *s* and *p* orbitals for O and H. These modifications are clearly indicated in Figure 4f and also in the corresponding figure legend in the revised manuscript; the corrections are shown below.

Fig. 4 Theoretical analysis to disclose the mechanisms for superior antioxidant-like activities. **f** PDOS (where Ru corresponds to the *d* orbital, and OOH is the superposition of *s* and *p* orbitals for O and H) and **g** electrostatic potential analysis of Ru-hydroxide and Ru-oxide with the adsorption of a *OOH intermediate.

Additionally, we have conducted an analysis of the PDOS for the Ru sites before and after the adsorption of OOH, which is presented in Supplementary Fig. 23. Our results indicate that the orbitals of Ru in Ru-hydroxide undergo alterations following the adsorption of OOH, suggesting a critical interaction between Ru and OOH. A similar trend was observed in Ru-oxide. It is noteworthy that the orbital overlap of Ru and OOH in Ru-hydroxide is less pronounced than that in Ru-oxide, indicating that the interaction between Ru and OOH in Ru-hydroxide is weaker, which facilitates the desorption of OOH. The corresponding details have been added to the revised manuscript and revised supplementary information, which are also shown below:

Page 12 in the revised manuscript: “The partial density of states (PDOS) peaks observed before and after the adsorption of *OOH indicate that the orbitals of Ru in Ru-hydroxide undergo alterations, suggesting a critical interaction between Ru and *OOH (Supplementary Fig. 23).”

Supplementary Fig. 23. Partial density of states (PDOS) of Ru-hydroxide and Ru-oxide **a** before and **b** after *OOH adsorption. Source data are provided as a Source Data file.

(7) *The pioneering strategy of mimicking intracellular antioxidant defense systems to fabricate antioxidative materials endowed with proton/electron transfer capabilities marks a significant conceptual leap. While the proton transfer pathway delineated in Figure 3 has been rigorously confirmed, more clarification is required to elucidate the electron transfer processes.*

Response to comment:

Thank you for your valuable feedback and support regarding our new approach to fabricating antioxidase-like materials by mimicking intracellular antioxidant defense systems with proton/electron transfer capabilities. While we have confirmed the proton transfer pathway with considerable rigor, we recognize the need for further clarification on the electron transfer processes. Theoretical calculations reveal that proton transfer is closely linked to electron transfer; each reaction step involving the Ru active site and oxygen species is a redox reaction characterized by the transfer of electrons (Supplementary Fig. 22). For instance, Hirshfeld charge analysis reveals that the electron density at the Ru site starts at $-0.675 |e|$ (**S0**) and changes to $-0.673 |e|$ (**S1**) during the decomposition of H_2O_2 , confirming that electron transfer occurs at the Ru site. This electron transfer continues throughout the reaction cycle, with the Ru site reverting to its initial state upon completion of the reaction. We believe this clarification will satisfactorily address your queries regarding the electron transfer processes involved in our study and enrich our manuscript. The corresponding details have been added to the revised manuscript and revised supplementary information, as also shown below:

Page 11 in the revised manuscript: “Furthermore, the Hirshfeld charge analysis illustrated in Supplementary Fig. 22 indicates that, aided by the presence of lattice hydrogen, the Ru center in Ru-hydroxide exhibits significant electron transfer and rapid valence transitions throughout the catalytic process.”

Supplementary Fig. 22. Hirshfeld charge analysis of Ru along the reaction pathway of Ru-hydroxide. Source data are provided as a Source Data file.

(8) Does the hydroxylation methodology present a scalable solution for the broader class of Ru-based materials?

Response to comment:

Thank you for your insightful comments regarding the broader application potential of the proposed hydroxylation methodology applied to the other Ru-based materials. We believe that the proposed method can be extended to many other Ru-based materials. During our preliminary experiments, we have synthesized several different types of Ru-doped metal layer double hydroxides and Ru-doped metal oxides, with a particular focus on the effects of surface H species (or OH species) on ROS catalytic performance. The synthesized materials include Ru-NiFe(OH)_x, Ru-NiFeO_x, Ru-CoFe(OH)_x, Ru-CoFeO_x, Ru-CoNi(OH)_x, and Ru-CoNiO_x (Fig. R4). Through rigorous experiments, we have confirmed that the catalase (CAT)-like activities of Ru-doped metal hydroxides consistently surpass that of their corresponding oxides (Fig. R5). This finding underscores that our proposed hydroxylation strategy is a universal method for designing Ru-based materials, which can enhance the ROS catalytic properties significantly. Given the superior ROS catalytic efficacy demonstrated by Ru-CoNi(OH)_x in this work, we focus on exploring this particular antioxidant-like material for detailed studies related to ROS elimination and maxillofacial bone regeneration. Its superior performance positions it as a promising candidate for further exploration in biocatalytic ROS elimination and anti-inflammatory applications. Moving forward, we plan to expand our research to investigate the surface hydroxylation strategy to a broader range of biocatalytic materials. This will allow us to investigate their biocatalytic properties in greater depth, ultimately contributing to a more comprehensive understanding of their potential applications in various fields. Thank you once again for your insightful suggestions.

Fig. R4. XRD patterns of **a** Ru-NiFe(OH)_x and Ru-NiFeO_x, **b** Ru-CoFe(OH)_x and Ru-CoFeO_x, and **c** Ru-CoNi(OH)_x and Ru-CoNiO_x.

Fig. R5. **a** H₂O₂ elimination ratio of different biocatalysts (n = 3 independent experiments, data are presented as mean ± SD). **b** The O₂ concentration produced by different biocatalysts was measured by an oxygen dissolving meter in the presence of H₂O₂ (n = 3 independent experiments, data are presented as mean ± SD).

(9) *In-situ* experimentation stands as a valuable technique for substantiating catalytic mechanisms; however, the authors are encouraged to delineate the specific procedural details of the *in-situ* assays.

Response to comment:

Thank you for your constructive suggestion regarding the need for procedural details of the *in-situ* assays. We apologize for not including this information in our initial manuscript. As you noted, *in-situ* Fourier transform infrared (FTIR) spectroscopy is indeed a powerful method to elucidate biocatalytic mechanisms. In our study, *in-situ* FTIR measurements were conducted using an infrared spectrometer (Thermo Scientific, iS50 FTIR) equipped with an *in-situ* spectrum cell (Shanghai Yuanfang Technology Co., Ltd., SPECEL-III). We typically prepare a Nafion solution consisting of 210 μL of isopropyl alcohol, 750 μL of deionized water, and 40 μL of Nafion (perfluorosulfonic acid ion exchange resin, Energy Chemical, 5% w/w in water and 1-propanol). This is followed by the

formation of a catalyst/Nafion solution at a concentration of 10 mg/mL. During the experiment, we deposited 40 μ L of the biocatalyst solution onto a ZnSe crystal surface, allowed it to dry, and then installed the crystal into the *in-situ* cell. Subsequently, we added 5 mL of a 0.5 M H₂O₂ solution to the *in-situ* spectrum cell, and we collected the *in-situ* FTIR spectra at specific intervals, maintaining a reaction time of 10 minutes. The corresponding details have been added to the revised supplementary information, as also shown below:

Page 33 in the revised supplementary materials: “**Details for *in-situ* Fourier transform infrared (FTIR) measurement.** In our study, *in-situ* FTIR measurements were conducted using an infrared spectrometer (Thermo Scientific, iS50 FTIR) equipped with an *in-situ* spectrum cell (Shanghai Yuanfang Technology Co., Ltd., SPECEL-III). We typically prepare a Nafion solution consisting of 210 μ L of isopropyl alcohol, 750 μ L of deionized water, and 40 μ L of Nafion (perfluorosulfonic acid ion exchange resin, Energy Chemical, 5% w/w in water and 1-propanol). This is followed by the formation of a biocatalyst/Nafion solution at a concentration of 10 mg/mL. During the experiment, we deposited 40 μ L of the biocatalyst solution onto a ZnSe crystal surface, allowed it to dry, and then installed the crystal into the *in-situ* cell. Subsequently, 5 mL of a 0.5 M H₂O₂ solution was added to the *in-situ* cell, and we collected the *in-situ* FTIR spectra at specific intervals, maintaining a reaction time of 10 minutes.”

(10) *The inclusion of unstained control data in the supplementary materials is imperative to corroborate the threshold selection in apoptosis flow cytometry analyses.*

Response to comment:

Thank you for your helpful suggestions. We acknowledge our oversight in not including unstained controls to support our threshold setting. In response to your recommendation, we have now included these controls in the supplementary materials to adequately support our choice of threshold. We appreciate your thorough review, and the relevant updates can be found in the revised supplementary materials, as also shown below:

Supplementary Fig. 34. Apoptosis analysis by flow cytometry of Annexin V-FITC/PI stained hMSCs. The blank group indicates no staining; FITC and PI groups are single-dye staining with FITC and PI, respectively (n = 3 independent replicates).

(11) *Figure 5 illustrates the beneficial effects of Ru-hydroxide on human mesenchymal stem cells (hMSCs) across a spectrum of cellular functions, inferred through KEGG pathway analysis in the context of H₂O₂ and Ru-hydroxide treatments. Nevertheless, the authors should anchor these inferences with pertinent literature to establish the nexus between the observed cellular functions and the implicated pathways.*

Response to comment:

Thank you for your constructive feedback and helpful recommendations. We fully agree with your advice and have included pertinent references to elucidate the relationship between the cellular functions observed in human mesenchymal stem cells (hMSCs) and the implicated pathways, as discussed in Figure 6. These additions enhance the validity of our findings and provide a comprehensive context for our interpretations. The corresponding references can be found in the revised manuscript, as also shown below:

Page 17 in the revised manuscript: “Then, the top-enriched KEGG pathways in the comparison between H₂O₂ and Ru-hydroxide treatments were analyzed (Fig. 6b). It speculates that the Ru-

hydroxide could defend the oxidative stress of hMSC in ROS-rich microenvironments through the FoxO signaling pathway⁶⁸ and HIF-1 signaling pathway⁶⁹. The positive impacts of Ru-hydroxide on hMSC are reflected by cell cytoskeleton and adhesion (regulation of actin cytoskeleton and focal adhesion⁷⁰), cellular DNA damage and function (p53 signaling pathway⁷¹), cellular activity (ECM-receptor interaction, cellular senescence, and cell cycle), and cell apoptosis (apoptosis)⁷¹. In addition, the Ru-hydroxide can protect stem cells by modulating inflammatory responses (mTOR signaling pathway⁷² and TNF signaling pathway⁷³), influencing stem cell development and matrix formation (signaling pathways regulating pluripotency of stem cells, ECM-receptor interaction, and MAPK signaling pathway⁷⁴), and subsequently regulating osteogenesis (Hippo signaling pathway⁷⁵, PI3K-Akt signaling pathway⁷⁶, and TGF-beta signaling pathway⁷⁷). The relevant top-enriched GO (Gene Ontology) terms also speculated similar cellular biological processes (Fig. 6c).”

(12) *The colour fill in figure 6g is different from the other figures, please standardise.*

Response to comment:

We appreciate your careful review and observation regarding the differing color fill in Figure 7g compared to the other figures. We have revised and standardized the color fill for Figure 7g to ensure consistency throughout. This revision has been incorporated into the revised manuscript and is presented below. We are grateful for your attention to detail and your valuable suggestions.

Fig. 7 Endogenous inflammation inhibition in LPS-induced ROS-associated mandibular defect by Ru-hydroxide. **c** Quantitative analysis of fluorescence mean intensity of TNF- α (n = 3 independent replicates), ### $p_{(ROSup)} = 0.0002$, $p_{(hydroxide-ROSup)} = 0.7954$, *** $p_{(Ru-hydroxide-ROSup)} < 0.0001$. **d** Mean intensity of IL-1 β (n = 3 independent replicates), ### $p_{(ROSup)} < 0.0001$, $p_{(hydroxide-ROSup)} = 0.2970$, *** $p_{(Ru-hydroxide-ROSup)} < 0.0001$. **g** Mean intensity of γ -H2A.X (n = 3 independent replicates), ### $p_{(ROSup)} < 0.0001$, $p_{(hydroxide-ROSup)} = 0.6664$, *** $p_{(Ru-hydroxide-ROSup)} < 0.0001$.

Reviewer #3 (Remarks to the Author):

“In this article, Wang and colleagues evaluated the use of novel ROS inhibitor/quencher called Ru-doped layered double hydroxide (Ru-hydroxide) for superior redox homeostasis and maxillofacial bone regeneration. Specifically, RU-hydroxide attenuates ROS in stem cells and decreases DNA damage and increases the stem cell potential. Further, Ru-hydroxide can effectively sustain stem cell viability, proliferation and osteogenic differentiation in elevated ROS environments. It also showed in-vivo biocatalytic microenvironment modulations during inflammatory bone tissue regeneration. Overall the manuscript is well executed, however, there are some queries and concerns authors need to address for the improvement/clarifications that supports the conclusions.”

Response to the general comment:

We sincerely appreciate your helpful comments and constructive suggestions on our manuscript. Your comments and suggestions have played a crucial role in enhancing the quality of our work. In light of your insights and other reviewers' comments, we have comprehensively revised the manuscript, ensuring that all queries and concerns are addressed in both the updated manuscript and the supplementary materials. Your guidance has been invaluable in refining our study, and we are confident that these revisions have significantly improved the paper. Thank you once again for your insightful contributions.

(1) While RU-Hydroxide shows to deplete ROS concentration in stem cells, how will it achieve the optimal concentration of ROS? Mainly because low level ROS is also important for stem cells to maintain quiescence and self-renewal (PMID: 24974178).

Response to comment:

Thank you for your insightful comments and helpful suggestions. We agree with your statement that low levels of ROS are crucial for stem cells to maintain quiescence and self-renewal. Specifically, normal intracellular ROS levels, such as those of H₂O₂, are very low. Intracellular H₂O₂ concentrations below 10 nM, as well as extracellular concentrations below 1 μM, favor cell proliferation, migration, and angiogenesis (*Adv. Cancer Res.* **2014**, 122, 1-67; *Redox Biol.*, **2017**, 11,

613-619; *Nat. Rev. Mol. Cell Biol.*, **2020**, 21, 363). In contrast, inflammatory extracellular H₂O₂ concentration can reach up to 100 μM, which is detrimental to stem cells, highlighting the necessity of removing excess ROS to restore the cellular microenvironments to normal levels (*Adv. Funct. Mater.*, **2020**, 30, 2001771; *Angew. Chem. Int. Ed.*, **2021**, 60, 7323-7332).

To investigate how Ru-hydroxide depletes ROS concentrations in stem cells while maintaining optimal ROS levels, we collected supernatants from cell culture experiments under various treatment conditions and treatment times to assess remaining ROS (mainly H₂O₂) concentrations. Our results indicate that Ru-hydroxide does not completely reduce H₂O₂ concentrations to zero. Instead, because the tissue environment and cell metabolism also produce H₂O₂, the MSCs+ H₂O₂ + Ru-hydroxide can maintain a relatively stable concentration of ~0.38 μM H₂O₂ after 12 h, which is below the 1 μM threshold and comparable to the level in the bare MSCs group without any treatment (Supplementary Fig. 25). This suggests that the Ru-hydroxide treatment may not adversely affect the H₂O₂-related self-renewal capability of the stem cells. Moreover, we propose that the low concentration of H₂O₂ observed is a result of the microenvironment reaching a dynamic equilibrium. We found that the reactivity of the biocatalyst is concentration-dependent; as the reaction proceeds and the H₂O₂ substrate concentration decreases, the reaction rate slows down. Given that tissue environments and cell metabolism constantly produce H₂O₂ (*Redox Biol.*, 2017, 11, 613-619; *Nat. Rev. Mol. Cell Biol.*, 2020, 21, 363), the H₂O₂ level of the Ru-hydroxide treated group can reach a dynamic equilibrium, which is essential for maintaining cell growth and development. This explains why our experiments indicate that the material does not inhibit stem cell renewal after ROS clearance; rather, it supports the regenerative function of stem cells, aligning with your observations in cellular experiments.

Regarding the real-time measurement of the other intracellular ROS levels *in vivo*, the complexity of the microenvironment and ultralow concentration, coupled with the short lifespan of ROS, poses significant challenges to real-time intracellular measurements. Currently, we have no suitable method to measure the other types of ROS; we will try to explore the other ROS once there are suitable methods in our future studies. The corresponding details have been added to the revised manuscript and revised supplementary materials, as also shown below:

Page 14 in the revised manuscript: “Upon exposure of hMSCs to H₂O₂ (100 μM), a significant increase in fluorescence intensity (green) indicative of high intracellular ROS levels is observed. Treatment with Ru-hydroxide led to a marked attenuation of the green fluorescence signals, whereas the control sample (Ru-oxide) exhibited only a slight decrease in the signal. And there is no significant reduction in ROS levels for the bare oxide and hydroxide groups. **Notably, because the cell metabolism also produces H₂O₂, the MSCs+ H₂O₂ + Ru-hydroxide can maintain a relatively stable concentration of ~0.38 μM H₂O₂, which is below the 1 μM threshold and comparable to the level in the bare MSCs group without any treatment. This suggests that the Ru-hydroxide treatment may not adversely affect the H₂O₂-related self-renewal capability of the stem cells (Supplementary Fig. 25)⁶⁷.”**

Standard curve for H₂O₂

Detecting H₂O₂ concentration in various cell culture environments

Supplementary Fig. 25. **a** Determination of absorbance at different H₂O₂ concentrations by horseradish peroxidase (HRP) and 3,3',5,5'-tetramethylbenzidine (TMB). **b** The standard curve for H₂O₂. **c** Absorbance curves for H₂O₂ detection in different cell culture environments after co-culture for 1, 4, 12 h. **d** H₂O₂ concentration in different cell culture environments. The results indicate that Ru-hydroxide does not reduce H₂O₂ concentrations to zero. Instead, because the cell metabolism also produces H₂O₂, the MSCs+ H₂O₂ + Ru-hydroxide can maintain a relatively stable concentration of ~0.38 μM H₂O₂, which is below the 1 μM threshold and comparable to the level in the bare MSCs group without any treatment. This suggests that the Ru-hydroxide treatment may not adversely affect

the H₂O₂-related self-renewal capability of the stem cells. Source data are provided as a Source Data file.

(2) Bone healing is a month to year long process. How stable is the Ru-hydroxide? will the new stem cells that arise at the end of bone reformation will have similar access to Ru-hydroxide in-vivo compared to the stem cells that forms early in the repair?

Response to comment:

Thank you for your thoughtful review and the insightful questions you raised regarding the stability of Ru-hydroxide and its accessibility at the end of bone reformation. You are absolutely right that bone healing is a long-term process, and its duration can vary significantly depending on factors, such as the species of the animal and the extent of the bone defects. In our study, we observed that the Ru-hydroxide treatment group was able to achieve significant repair of mandible defects within approximately two months. To investigate how long the defects and stem cells can access Ru-hydroxide, we have conducted *in vitro* experiments to evaluate the degradation performances of the materials. Specifically, we have immersed 1 mg of Ru-hydroxide in 10 mL of phosphate-buffered saline (PBS) and cell culture medium, respectively. The results from our inductively coupled plasma mass spectrometry (ICP-MS) analysis indicated that Ru-hydroxide degrades gradually, with cobalt (Co) ions showing about 50% degradation after four weeks in PBS and approximately 70% degradation after four weeks in cell culture medium (Supplementary Fig. 39). This degradation pattern suggests that Ru-hydroxide primarily influences early oxidative stress and inflammatory cells for tissue regeneration, which will have gradually decreased impact on those involved in later stages of bone reconstruction.

In general, inflammation mainly occurs in the early stage of inflammatory bone defect repair. In this work, after the addition of antioxidase-like materials, the severe inflammation could be efficiently controlled. Meanwhile, in the later stage of bone repair, it is believed that the antioxidase-like materials may not be needed since the inflammation is very weak and even disappears. Additionally, as discussed in our response to your first question, two months after the bone reconstruction, the redox microenvironment may have already achieved stable status, causing a

significant drop in the H₂O₂ levels and also other types of ROS. However, if inflammation occurs again, we can take a secondary injection of antioxidase-like materials to inhibit the emerged inflammatory microenvironments. We acknowledge that these conjectures warrant further investigation in the future, and we will develop suitable tracking methodologies in future studies to gain deeper insights into the behaviors and interactions of these materials during the long-term tissue generation process. Again, we thank you for your valuable suggestions, which will undoubtedly guide our future research. The corresponding details have been added to the revised manuscript and revised supplementary materials, as also shown below:

Page 23 in the revised manuscript: “Additionally, the observation that Ru-hydroxide undergoes degradation during the treatment cycle indicates its efficient impact on the early stage of bone reconstruction and limited effects on the long-term generation process (Supplementary Fig. 39).”

Degradation of Ru-hydroxide in PBS

Degradation of Ru-hydroxide in cell culture medium

Supplementary Fig. 39. Inductively coupled plasma mass spectrometry (ICP-MS) was used to measure **a** the concentration of metal ions released and **b** the percentage of material degradation at different time points. 1 mg of the Ru-hydroxide was immersed in 10 mL phosphate buffer saline (PBS) to explore the degradation performance of the material over time *in vitro*. **c** The concentration of metal ions released and **d** the percentage of material degradation at different time points; 1 mg of the Ru-hydroxide was immersed in 10 mL cell culture medium (HUXMX-90021, Cyagen, China). Source data are provided as a Source Data file.

(3) What happens to non-stem cells that picks up Ru-hydroxide?

Response to comment:

We sincerely appreciate your valuable feedback, which has allowed us to improve the quality of this paper. In addition to stem cells, osteoblasts and dental pulp cells are also present in the damaged tissue surrounding tooth roots, and the retention of these cells benefits tissue repair. To detect the interaction of these non-stem cells with Ru-hydroxide, we employed 2,7-dichlorodihydrofluorescein diacetate (DCFH-DA) as a fluorescence probe to assess intracellular ROS levels (Supplementary Figs. 26, 27). Following treatment of MC3T3 (a pe-osteoblast cell line) or human dental pulp cells (hDPCs) with H₂O₂ (100 μM), a pronounced green fluorescence is observed, indicating elevated intracellular ROS levels. Notably, the green fluorescence signals are significantly diminished after treatment with Ru-hydroxide; meanwhile, the cell morphology and cell number are similar to those of the control group. These findings suggest that Ru-hydroxide is biocompatible to non-stem cells, which also holds great promise for applications in removing ROS and protecting MC3T3 or hDPCs from oxidative stress. The corresponding details can be found in the revised manuscript and revised supplementary materials, as also shown below:

Page 14 in the revised manuscript: “Treatment with Ru-hydroxide led to a marked attenuation of the green fluorescence signals, whereas the control sample (Ru-oxide) exhibited only a slight decrease in the signal. And there is no significant reduction in ROS levels for the bare oxide and hydroxide groups. Notably, because the cell metabolism also produces H₂O₂, the MSCs+ H₂O₂ + Ru-hydroxide can maintain a relatively stable concentration of ~0.38 μM H₂O₂, which is below the 1 μM threshold

and comparable to the level in the bare MSCs group without any treatment. This suggests that the Ru-hydroxide treatment may not adversely affect the H₂O₂-related self-renewal capability of the stem cells (Supplementary Fig. 25)⁶⁷. In addition to stem cells, osteoblasts and dental pulp cells are also present in the tissue surrounding the damaged tooth root, and the retention of these cells benefits tissue repair. As illustrated in Supplementary Figs. 26, 27, Ru-hydroxide also shows efficient defense capability on ROS damage in MC3T3 (a pre-osteoblast cell line) and hDPCs.”

Supplementary Fig. 26. **a** Fluorescence images and **b** mean fluorescence intensity of DCFH-DA staining (n = 3 independent replicates), ### $p_{(H_2O_2)} < 0.0001$, *** $p_{(Ru-hydroxide+H_2O_2)} < 0.0001$. Data are presented as means \pm SD., ### $P < 0.001$, *** $P < 0.001$; one-way ANOVA with multiple comparisons test. Scale bar: 50 μ m. Ctrl (MC3T3+PBS), MC3T3 indicates a pre-osteoblast cell line. Source data are provided as a Source Data file.

Supplementary Fig. 27. a Fluorescence images and **b** mean fluorescence intensity of DCFH-DA staining (n = 3 independent replicates), ### $p_{(H_2O_2)} < 0.0001$, *** $p_{(Ru-hydroxide+H_2O_2)} < 0.0001$. Data are presented as means \pm SD., ### $P < 0.001$, *** $P < 0.001$; statistical significance was calculated using one-way ANOVA followed by Tukey's post-hoc test for multiple comparisons, all tests were two-sided. Scale bar: 50 μ m. Ctrl (hDPCs+PBS). Source data are provided as a Source Data file.

(4) *Gamma-H2AX measures indirect double strand breaks and is highly variable based on the cell type or the time we fix these cells. For a more prominent study like this, it is important that the authors evaluate these DNA damage responses using more relevant assays like alkaline and neutral COMET assays for single and double strand breaks, oxidative DNA damage/8-Oxoguanine, other signal proteins modified due to free radicals and/or Chromosomal aberrations.*

Response to comment:

We sincerely appreciate your helpful and constructive comments aimed at enhancing the quality of our manuscript. We acknowledge the insufficient data in our research concerning the DNA damage responses of cells with the treatment of Ru-hydroxide in high ROS microenvironment. As suggested by your comments, we have added new evaluations on the DNA damage responses using more relevant assays like alkaline and neutral comet assays for single and double-strand breaks (Supplementary Fig. 30). Compared to H₂O₂ treatment, comet assays revealed a significant decrease in DNA damage with Ru-hydroxide treatment. Then, we have added the analysis of the oxidative DNA damage/8-Oxoguanine by immunofluorescence staining. The resulting data indicate that the Ru-hydroxide group shows a less 8-Oxoguanine (8-oxoG) compared to the other experimental groups, suggesting its efficacy in protecting hMSCs from the oxidative DNA damage (Supplementary Fig. 29). Furthermore, we referred to some literatures and found that H₂O₂ treatment could increase p-ERK signaling protein (*Mol. Neurobiol.*, **2023**, 60, 5725-5737; *Free Radic. Biol. Med.*, **2021**, 166, 265-276) and p-ERK signaling protein could mediate DNA damage signaling pathways (*Mol. Biol. Rep.*, **2012**, 39, 8007-8014). So, we have included an analysis of p-ERK by immunofluorescence staining (Supplementary Fig. 32). The resulting data indicate that H₂O₂ treatment could increase p-ERK signaling protein, and the Ru-hydroxide group shows a decrease in

p-ERK signaling protein expression. These findings offer deeper insights into the DNA damage responses of cells with the treatment of Ru-hydroxide in high ROS microenvironment. We appreciate your thoughtful input, which has significantly informed our work. The corresponding details can be found in the revised manuscript and revised supplementary information, as also shown below:

Page 17 in the revised manuscript: “Furthermore, GSEA is carried out to reveal the gene signatures of a series of biological processes (Figs. 6d, e). Compared to the H₂O₂ group, the Ru-hydroxide group shows the reduction of ‘Intrinsic apoptotic signaling pathway’ and ‘Apoptosis-multiple species’, suggesting its stem cell protection activity by reducing apoptosis induced by oxidative stress. It has been recognized that excessive ROS levels can cause an irreversible attack on DNA⁷⁸. Thereafter, we systematically explore the catalytic therapeutic potential of Ru-hydroxide in reversing ROS-induced DNA damage and apoptosis. Initially, we assess the phosphorylation of γ -H2A.X, a marker for double-stranded DNA breaks. As the immunofluorescence staining and the spectral analysis of nuclei reveal obvious γ -H2A.X signals co-express with Dapi in the H₂O₂ group (Fig. 6f), while the Ru-hydroxide group shows significantly reduced signals, which is as low as the Ctrl group as indicated by quantitative analysis in Fig. 6i. Additionally, as depicted in Fig. 6g and **Supplementary Fig. 29**, compared to the Ctrl group, the signals of DNA/RNA damage **and the oxidative DNA damage (8-Oxoguanine)** markers in nuclei are significantly elevated in the H₂O₂ group, while the Ru-hydroxide group display substantially reduced red signals, resembling the fluorescence levels in the Ctrl group (Figs. 6h, j). **Furthermore, we have evaluated DNA damage responses using more relevant assays like alkaline and neutral comet assays for single and double-strand breaks (Supplementary Fig. 30). Compared to H₂O₂ treatment, comet assays reveal a significant decrease in DNA damage with Ru-hydroxide treatment. p-ATM and p-ERK were known to participate in ROS-induced DNA damage response⁷⁹⁻⁸¹. Thereafter, we perform an analysis of p-ATM and p-ERK by immunofluorescence staining. The resulting data indicate that H₂O₂ treatment could increase these protein expressions, while the Ru-hydroxide group shows significantly reduced signals, which is as low as the Ctrl group, as indicated by quantitative analysis in **Supplementary Figs. 31, 32**. Besides, flow cytometry analysis categorized cell populations into early apoptotic and late apoptotic stages (Figs. 6l, m and **Supplementary Figs. 33, 34**). The cells treated with H₂O₂ exhibit an obvious apoptosis ratio, while the Ru-hydroxide group displays a similar ratio to the Ctrl group, indicating its efficient protection**

against oxidative stress-induced cell apoptosis, thus maintaining good cellular redox balance as the Ctrl group (Fig. 6k).”

Supplementary Fig. 29. **a** Fluorescence images and **b** mean fluorescence intensity of 8-Oxoguanine (8-oxoG) staining (n = 30 independent replicates), $###p_{(H_2O_2)} < 0.0001$, $***p_{(Ru-oxide+H_2O_2)} < 0.0001$, $***p_{(Ru-hydroxide+H_2O_2)} < 0.0001$. Data are presented as means \pm SD., $###P < 0.001$, $***P < 0.001$; statistical significance was calculated using one-way ANOVA followed by Tukey’s post-hoc test for multiple comparisons, all tests were two-sided. Scale bar: 50 μ m. Source data are provided as a Source Data file.

Supplementary Fig. 30. **a** Neutral and alkaline comet analysis. **b-e** Tail-moment and % tail DNA in hMSCs (n = 60 independent replicates). In **b**, $**p_{(H_2O_2+Ru-hydroxide)} = 0.0016$. In **c**, $**p_{(H_2O_2+Ru-hydroxide)} = 0.0094$. In **d**, $***p_{(H_2O_2+Ru-hydroxide)} < 0.0001$. In **e**, $***p_{(H_2O_2+Ru-hydroxide)} < 0.0001$. Data are presented as means \pm SD., $**P < 0.01$, $***P < 0.001$. Statistical significance was calculated using two-tailed

Student's t-test, all tests were two-sided. Scale bar: 100 μ m. Source data are provided as a Source Data file.

Supplementary Fig. 32. **a** Fluorescence images and **b** mean fluorescence intensity of p-ERK (phosphorylated extracellular signal-regulated kinases) staining ($n = 30$ independent replicates), $\#\#p_{(H_2O_2)} = 0.0045$, $\#\#p_{(Ru-hydroxide+H_2O_2)} = 0.0024$. Data are presented as means \pm SD., $\#\#P < 0.01$, $\#\#P < 0.01$; statistical significance was calculated using one-way ANOVA followed by Tukey's post-hoc test for multiple comparisons, all tests were two-sided. Scale bar: 50 μ m. Source data are provided as a Source Data file.

(5) Figure 6B: Treatment with Ru-hydroxide reduces the levels of TNF-alpha and IL-1B well beyond the control levels. Again, low levels of TNF-alpha and IL-1B is also required to maintain the normal cellular homeostasis. Especially, TNF-alpha is required for normal hematopoiesis and will Ru-hydroxide impact the normal hematopoiesis in maxillofacial regions?

Response to comment:

Thank you for your valuable comments regarding the need for additional consideration of the levels of TNF-alpha. Exactly as the point of the reviewer, we have evaluated the TNF-alpha expression in normal periapical tissue by immunofluorescence staining in Fig. R7; the data indicates that TNF-alpha also exists in normal tissue besides the inflammatory tissue. Therefore, we have

made a detailed comparison of the TNF- α concentration of the normal group and the Ru-hydroxide-ROSup group. It is found that the Ru-hydroxide-ROSup group displays higher TNF- α expression, suggesting that there will be sufficient TNF- α for the bone tissue reconstruction process and the Ru-hydroxide may not impact the normal hematopoiesis in maxillofacial regions. Additionally, our results demonstrate that while TNF- α and IL-1 β levels in our Ru-hydroxide-ROSup group were lower than those in the control group, it is important to clarify that the control group here represents mandible defects treated with saline; thus, the TNF- α concentration in the control group should be higher than in normal tissues. The above analysis suggests that the treatment may not compromise hematopoietic function. Once again, thank you for your valuable feedback, which has helped us enhance the clarity of our findings.

Fig. R7. a Fluorescence images of TNF- α and **b** quantitative analysis of fluorescence mean intensity of TNF- α ($n = 3$ independent replicates), $p = 0.0002$. Data are presented as means \pm SD., *** $P < 0.001$. Statistical significance was calculated using two-tailed Student's t-test; all tests were two-sided. Scale bar: 200 μ m.

(6) Most of the DNA damage response pathways were analyzed using genomic studies in the current study. However, most of the DNA damage responses are modified post-transcriptionally at the protein levels. Analysis of some of these proteins and their posttranslational modifications are as important as the genomic data about DNA damage responses as the authors have described in the article.

Response to comment:

Thanks for your important and helpful comments on improving the quality of our manuscript. As suggested by the reviewer, in the revised manuscript, we have evaluated some proteins that are related to the DNA damage response pathways and their posttranslational modifications. In KEGG pathway analysis between H₂O₂ and Ru-hydroxide+H₂O₂, p53 and MAPK signaling pathways shows enrichment (Fig. 6b). Furthermore, p-ATM (associated with p53 signaling pathway) and p-ERK (associated with MAPK signaling pathway) were known to participate in ROS-induced DNA damage response (*Nature*. **2022**, 604, 714-722; *Free Radic. Biol. Med.*, **2021**, 166, 265-276; *Mol. Biol. Rep.*, **2012**, 39, 8007-8014). So, we have added an analysis of p-ATM and p-ERK by immunofluorescence staining. The resulting data indicate that H₂O₂ treatment could increase these proteins' expression, while the Ru-hydroxide group shows significantly reduced signals, which is as low as the Ctrl group, as indicated by quantitative analysis in Supplementary Figs. 31, 32.

Page 17 in the revised manuscript: “Furthermore, GSEA is carried out to reveal the gene signatures of a series of biological processes (Figs. 6d, e). Compared to the H₂O₂ group, the Ru-hydroxide group shows the reduction of ‘Intrinsic apoptotic signaling pathway’ and ‘Apoptosis-multiple species’, suggesting its stem cell protection activity by reducing apoptosis induced by oxidative stress. It has been recognized that excessive ROS levels can cause an irreversible attack on DNA⁷⁸. Thereafter, we systematically explore the catalytic therapeutic potential of Ru-hydroxide in reversing ROS-induced DNA damage and apoptosis. Initially, we assess the phosphorylating of γ -H2A.X, a marker for double-stranded DNA breaks. As the immunofluorescence staining and the spectral analysis of nuclei reveal obvious γ -H2A.X signals co-express with Dapi in the H₂O₂ group (Fig. 6f), while the Ru-hydroxide group shows significantly reduced signals, which is as low as the Ctrl group as indicated by quantitative analysis in Fig. 6i. Additionally, as depicted in Fig. 6g and **Supplementary Fig. 29**,

compared the Ctrl group, the signals of DNA/RNA damage and the oxidative DNA damage (8-Oxoguanine) markers in nuclei are significantly elevated in the H₂O₂ group, while the Ru-hydroxide group display substantially reduced red signals, resembling the fluorescence levels in the Ctrl group (Figs. 6h, j). Furthermore, we have evaluated DNA damage responses using more relevant assays like alkaline and neutral comet assays for single and double-strand breaks (Supplementary Fig. 30). Compared to H₂O₂ treatment, comet assays reveal a significant decrease in DNA damage with Ru-hydroxide treatment. p-ATM and p-ERK were known to participate in ROS-induced DNA damage response⁷⁹⁻⁸¹. Thereafter, we perform an analysis of p-ATM and p-ERK by immunofluorescence staining. The resulting data indicate that H₂O₂ treatment could increase these protein expressions, while the Ru-hydroxide group shows significantly reduced signals, which is as low as the Ctrl group, as indicated by quantitative analysis in Supplementary Figs. 31, 32. Besides, flow cytometry analysis categorized cell populations into early apoptotic and late apoptotic stages (Figs. 6l, m and Supplementary Figs. 33, 34). The cells treated with H₂O₂ exhibit an obvious apoptosis ratio, while the Ru-hydroxide group displays a similar ratio to the Ctrl group, indicating its efficient protection against oxidative stress-induced cell apoptosis, thus maintaining good cellular redox balance as the Ctrl group (Fig. 6k).”

Supplementary Fig. 31. a Fluorescence images and **b** mean fluorescence intensity of p-ATM (phosphorylated-ataxia telangiectasia-mutated) staining (n = 30 independent replicates), ### $p_{(H_2O_2)} < 0.0001$, *** $p_{(Ru-hydroxide+H_2O_2)} < 0.0001$. Data are presented as means \pm SD., ### $P < 0.001$, *** $P < 0.001$; statistical significance was calculated using one-way ANOVA followed by Tukey’s post-hoc test for

multiple comparisons, all tests were two-sided. Scale bar: 50 μm . Source data are provided as a Source Data file.

Supplementary Fig. 32. **a** Fluorescence images and **b** mean fluorescence intensity of p-ERK (phosphorylated extracellular signal-regulated kinases) staining ($n = 30$ independent replicates), $##p_{(\text{H}_2\text{O}_2)} = 0.0045$, $**p_{(\text{Ru-hydroxide}+\text{H}_2\text{O}_2)} = 0.0024$. Data are presented as means \pm SD., $##P < 0.01$, $**P < 0.01$; statistical significance was calculated using one-way ANOVA followed by Tukey's post-hoc test for multiple comparisons, all tests were two-sided. Scale bar: 50 μm . Source data are provided as a Source Data file.

We thank all referees again for their helpful comments and important suggestions, and we hope that this significantly revised manuscript is now acceptable for publication in *Nature Communications*.

Best Regards,

Yours Sincerely,

Prof. Dr. Chong Cheng (on behalf of the authors)